# LEARNING VALUE FUNCTIONS FROM UNDIRECTED STATE-ONLY EXPERIENCE

**Matthew Chang**[*]  **Arjun Gupta**[*]  **Saurabh Gupta**
University of Illinois at Urbana-Champaign
{mc48, arjung2, saurabhg}@illinois.edu

## ABSTRACT

This paper tackles the problem of learning value functions from undirected state-only experience (state transitions without action labels *i.e.* $(s, s', r)$ tuples). We first theoretically characterize the applicability of Q-learning in this setting. We show that tabular Q-learning in discrete Markov decision processes (MDPs) learns the same value function under any arbitrary refinement of the action space. This theoretical result motivates the design of Latent Action Q-learning or LAQ, an offline RL method that can learn effective value functions from state-only experience. Latent Action Q-learning (LAQ) learns value functions using Q-learning on discrete latent actions obtained through a latent-variable future prediction model. We show that LAQ can recover value functions that have high correlation with value functions learned using ground truth actions. Value functions learned using LAQ lead to sample efficient acquisition of goal-directed behavior, can be used with domain-specific low-level controllers, and facilitate transfer across embodiments. Our experiments in 5 environments ranging from 2D grid world to 3D visual navigation in realistic environments demonstrate the benefits of LAQ over simpler alternatives, imitation learning oracles, and competing methods.

## 1 INTRODUCTION

Offline or batch reinforcement learning focuses on learning goal-directed behavior from pre-recorded data of undirected experience in the form of $(s_t, a_t, s_{t+1}, r_t)$ quadruples. However, in many realistic applications, action information is not naturally available (*e.g.* when learning from video demonstrations), or worse still, isn't even well-defined (*e.g.* when learning from the experience of an agent with a different embodiment). Motivated by such use cases, this paper studies if, and how, intelligent behavior can be derived from undirected streams of observations: $(s_t, s_{t+1}, r_t)$.[1]

At the face of it, it might seem that observation-only data would be useless towards learning goal-directed policies. After all, to learn such a policy, we need to know what actions to execute. Our key conceptual insight is that while an observation-only dataset doesn't tell us the precise action to execute, *i.e.* the policy $\pi(a|s)$; it may still tell us which states are more likely to lead us to the goal than not, *i.e.* the value function $V(s)$. For example, simply by looking at someone working in the kitchen, we can infer that approaching the microwave handle is more useful (*i.e.* has higher value) for opening the microwave than to approach the buttons. Thus, we can still make use of observation-only data, if we focused on learning value functions as opposed to directly learning goal-directed policies. Once we have learned a good value function, it can be used to quickly acquire or infer behavior. Using learned value functions as dense rewards can lead to quick policy learning through some small amount of interaction in the environment. Alternatively, they could be used to directly guide the behavior of low-level controllers that may already be available for the agent (as is often the case in robotics) without any further training. Furthermore, decoupling the learning of value functions from policy learning enables deriving behavior for agents with a different embodiment as long as the overall solution strategy remains similar. Thus, the central technical question is how to learn a good value function from undirected observation streams. Is it even possible? If so, under what conditions? This paper tackles these questions from a theoretical and practical perspective.

---

[*] denotes equal contribution. Project website: https://matthewchang.github.io/latent_action_qlearning_site/.
[1] We assume $r_t$ is observed. Reward can often be sparsely labeled in observation streams with low effort.

We start out by characterizing the behavior of tabular Q-learning from Watkins (1989) under missing action labels. We note that Q-learning with naively imputed action labels is equivalent to the TD(0) policy evaluation, which serves as a simple baseline method for deriving a value function. However, depending on the policy that generated the data, the learned values (without any action grounding) can differ from the optimal values. Furthermore, it is possible to construct simple environments where the behavior implied by the learned value function is also sub-optimal.

Next, we present a more optimistic result. There are settings in which Q-learning can recover the optimal value function even in the absence of the knowledge of underlying actions. Concretely, we prove that if we are able to obtain an action space which is a strict refinement of the original action space, then Q-learning in this refined action space recovers the optimal value function.

This motivates a practical algorithm for learning value functions from the given undirected observation-only experience. We design a latent-variable future prediction model that seeks to obtain a refined action space. It operates by predicting $s_{t+1}$ from $s_t$ and a discrete latent variable $\hat{a}$ from a set of actions $\hat{\mathbf{A}}$ (Section 4.1). Training this latent variable model assigns a discrete action $\hat{a}_t$ to each $(s_t, s_{t+1})$ tuple. This allows us to employ Q-learning to learn good value functions (Section 4.2). The learned value function is used to derive behavior (Section 4.3) either through some online interaction with the environment, or through the use of domain specific low-level controllers.

The use of a latent action space for Q-learning allows us to exceed the performance of methods based on policy evaluation (Edwards & Isbell, 2019), which will learn the value of the demonstration policy, not the optimal value function. Additionally, it side-steps the problem of reconstructing high-dimensional images faced by other state-only value learning methods (Edwards et al., 2020). Other approaches for learning from state-only data rely on imitating the demonstration data, which renders them unable to improve on sub-optimal demonstration data. See Section 7 for more discussion.

Our experiments in five environments (2D grid world, 2D continuous control, Atari game Freeway, robotic manipulation, and visual navigation in realistic 3D environments) test our proposed ideas. Our method approximates a refinement of the latent space better than clustering alternatives, and in turn, learns value functions highly correlated with ground truth. Good value functions in-turn lead to sample efficient acquisition of behavior, leading to significant improvement over learning with only environment rewards. Our method compares well against existing methods that learn from undirected observation-only data, while being also applicable to the case of high-dimensional observation spaces in the form of RGB images. We are also able to outperform imitation learning methods, even when these imitation learning methods have access to privileged ground-truth action information. Furthermore, our method is able to use observation-only experience from one agent to speed up learning for another agent with a different embodiment.

## 2 PRELIMINARIES

Following the notation from Sutton & Barto (2018), our Markov decision process (MDP) is specified by $(\mathbf{S}, \mathbf{A}, p, \gamma)$, where $\mathbf{S}$ is a state space, $\mathbf{A}$ is an action space, $\gamma$ is the discount factor, and $p(s', r|s, a)$ is the state/reward joint dynamics function. It specifies the probability distribution that the agent ends up in state $s'$, receives a reward of $r$ on executing action $a$ from state $s$.

Offline or batch RL (Lange et al., 2012; Levine et al., 2020) studies the problem of deriving high reward behavior when only given a dataset of experience in an MDP, in the form of a collection of quadruples $(s, a, s', r)$. In this paper, we tackle a harder version of this problem where instead we are only given a collection of triplets $(s, s', r)$, *i.e.* experience without information about intervening actions. In general, this dataset could contain any quality of behavior. In contrast to some methods (see Section 7), we will not assume that demonstrations in the dataset are of high quality, and design our method to be robust to sub-optimal data. Using such a dataset, our goal is to learn good value functions. A value function under a policy $\pi$ is a function of states that estimates the expected return when starting in $s$ and following $\pi$ thereafter.

In this paper, we will focus on methods based on Q-learning (Watkins, 1989) for tackling this problem. Q-learning has the advantage of being *off-policy*, *i.e.*, experience from another policy (or task) can be used to learn or improve a different policy for a different task. Q-learning seeks to learn the optimal Q-function $Q^*(s, a)$ by iteratively updating $Q(s_t, a_t)$ to the Bellman equation. This process converges to the $Q^*$ under mild conditions in many settings (Watkins, 1989).

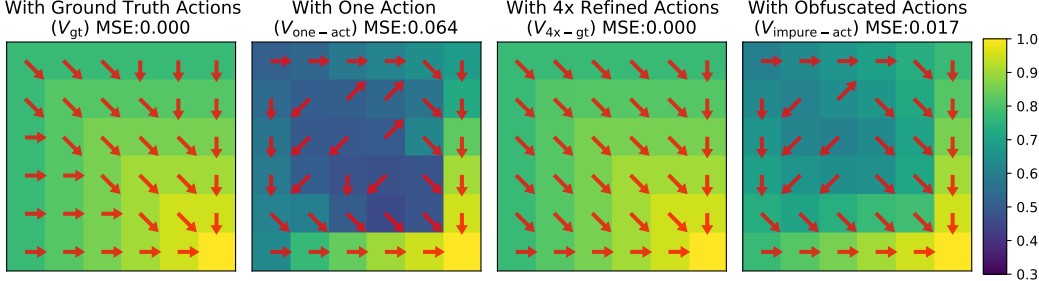

**Figure 1:** We visualize the learned value function when using different action labels for Q-learning: ground truth actions, one single action, a $4\times$ refined action space, and obfuscated actions. We also report the mean squared error from the optimal value function. Arrows show the behavior induced by the value function (picking neighboring state with highest value). We note that a) ignoring the intervening actions works poorly both in terms of value function estimates and the induced behavior, b) refined actions don't cause any degradation in performance, c) noisy actions that don't adhere the refinement constraint cause degradation in performance. See Section 3.2 for more details.

# 3 CHARACTERIZING Q-LEARNING WITHOUT TRUE ACTION LABELS

We characterize the outcome of Q-learning in settings where we don't have ground truth intervening actions in the offline dataset being used for Q-learning. Without action labels, one could simply assign all transitions the same label. In this case, Q-Learning becomes TD(0) policy evaluation. The induced value function isn't the optimal value function for the MDP, but rather the value according to the policy that generated the dataset. Depending on the dataset, this could be sub-optimal.

Next, we study if labeling $(s, s', r)$ samples with actions from a different action space $\hat{\mathbf{A}}$ to construct a new MDP could aid learning. More specifically, can the optimal Q-function for this new MDP, as obtained through Q-learning on samples $(s, s', r)$ labeled with actions from $\hat{\mathbf{A}}$, be useful in the original MDP? We show that under the right conditions the value function learned under the altered action space $\hat{\mathbf{A}}$ is identical to the value function learned for the original MDP.

## 3.1 OPTIMALITY OF ACTION REFINEMENT

Assume we have a Markov Decision Process (MDP) $M$ specified by $(\mathbf{S}, \mathbf{A}, p, \gamma)$. Let the action space $\mathbf{A}$ be composed of actions $a_1, a_2, a_3, ..., a_n \in \mathbf{A}$. We are interested in the value learned under a modified MDP, $\hat{M}$ composed of $(\mathbf{S}, \hat{\mathbf{A}}, \hat{p}, \gamma)$. We will show that if the actions and transitions $\hat{\mathbf{A}}$ and $\hat{p}$ are a *refinement* of $\mathbf{A}$ and $p$, then the value function learned on $\hat{M}$, $V_{\hat{M}}$ is identical to the value function learned on $M$, $V_M$. We define actions and transitions in $\hat{M}$ to be a refinement of those in $M$ when, a) in each state, for every action in $\hat{\mathbf{A}}$, there is at least one action in $\mathbf{A}$ which is functionally identical in the same state, and b) in each state, for each action in $\mathbf{A}$ is represented by at least one action in $\hat{\mathbf{A}}$ in that state.

**Definition 3.1** Given a discrete finite MDP, $M$ specified by $(\mathbf{S}, \mathbf{A}, p)$ and MDP, $\hat{M}$ specified by $(\mathbf{S}, \hat{\mathbf{A}}, \hat{p})$, $\hat{M}$ is a *refinement* of $M$ when

$$\underset{\hat{a} \in \hat{\mathbf{A}}, s \in \mathbf{S}}{\forall} \underset{a \in \mathbf{A}}{\exists} \underset{s', r}{\forall} \hat{p}(s', r | s, \hat{a}) = p(s', r | s, a), \text{ and } \underset{a \in \mathbf{A}, s \in \mathbf{S}}{\forall} \underset{\hat{a} \in \hat{\mathbf{A}}}{\exists} \underset{s', r}{\forall} \hat{p}(s', r | s, \hat{a}) = p(s', r | s, a),$$

Note that this definition of refinement requires a *state conditioned* correspondence between action behavior. Actions do not need to have to correspond across states.

**Theorem 3.1.** *Given a discrete finite MDP, $\hat{M}$ which is a refinement of $\hat{M}$ (Definition 3.1) then both MDPs induce the same optimal value function, i.e. $\forall_s V^*_{\hat{M}}(s) = V^*_M(s)$.*

We prove this by showing that optimal policies under both MDPs induce the same value function.

**Lemma 3.2.** *For any policy $\pi_M$ on $M$, there exists a policy $\pi_{\hat{M}}$ on $\hat{M}$ such that $V^{\pi_{\hat{M}}}_{\hat{M}}(s) = V^{\pi_M}_M(s)$, $\forall s$, and for any policy $\pi_{\hat{M}}$ on $\hat{M}$ there exists a policy $\pi_M$ on $M$ such that $V^{\pi_{\hat{M}}}_{\hat{M}}(s) = V^{\pi_M}_M(s)$ $\forall s$.*

**Figure 2: Approach Overview.** Our proposed approach Latent Action Q-Learning (LAQ) starts with a dataset of $(s, s', r)$ triples. Using the latent action learning process, each sample is assigned a latent action $\hat{a}$. Q-learning on the dataset of quadruples produces a value function, $\hat{V}(s)$. Behaviors are derived from the value function through densified RL, or by guiding low-level controllers.

For this lemma we introduce the notion of *fundamental actions*, which are actions which correspond to sets of actions which have the same state and reward transition distributions in a given state. We utilize the equivalence of fundamental actions between MDPs to construct a policy in the new MDP which induces the same value function as a given policy in the original MDP. We provide proofs for Theorem 3.1 and Lemma 3.2 in Section A.2.

### 3.2 GRIDWORLD CASE STUDY

We validate these results in a tabular grid world setting. In particular, we measure the error in learned value functions and the induced behavior, when conducting Q-learning with datasets with different qualities of intervening actions. The agent needs to navigate from the top left of a $6 \times 6$ grid to the bottom right with sparse reward. We generate data from a fixed, sub-optimal policy to evaluate all methods in an offline fashion (additional details in Section A.7). We generate 20K episodes with this policy, and obtain value functions using Q-learning under the following 4 choices for the intervening actions: **(1)** Ground truth actions ($V_{\text{gt}}$), **(2)** One action ($V_{\text{one-act}}$, ammounts to TD(0) policy evaluation), **(3)** $4\times$ refinement of original action space ($V_{4\times\text{-gt}}$). We modify the data so that each sample for a particular action in the original action space is randomly mapped to one of 4 actions in the augmented space. **(4)** Obfuscated actions ($V_{\text{impure-act}}$). Original action with probability 0.5, and a random action with probability 0.5.

Figure 1 shows the learned value functions under these different action labels, and reports the MSE from the true value function, along with induced behavior. In line with our expectations, $V_{4\times\text{-gt}}$ which uses a refinement of the actions is able to recover the optimal value function. $V_{\text{one-act}}$ fails to recover the optimal value function, and recovers the value corresponding to the policy that generated the data. $V_{\text{impure-act}}$, under noise in action labels (non-refinement) also fails to recover the optimal value function. Furthermore, the behavior implied by $V_{\text{impure-act}}$ and $V_{\text{one-act}}$ is sub-optimal. We also analyze the effect of the action impurity on learned values and implied behavior. Behavior becomes increasingly inaccurate as action impurity increases. More details in Section A.4.

## 4 LATENT ACTION Q-LEARNING

Our analysis in Section 3 motivates the design of our approach for learning behaviors from state-only experience. Our proposed approach decouples learning into three steps: mining *latent* actions from state-only trajectories, using these latent actions for Q-learning to obtain value functions, and learning a policy to act according to the learned value function. As per our analysis, if learned latent actions are a *state-conditioned refinement* of the original actions, Q-learning will result in good value functions, that will lead to good behaviors. Refer to Algorithm 1 for details.

### 4.1 LATENT ACTIONS FROM FUTURE PREDICTION

Given a dataset $\mathbf{D}$ of observations streams $\ldots, o_t, o_{t+1}, \ldots$, the goal in this step is to learn *latent* actions that are a refinement of the actual actions that the agent executed [2]. We learn these latent actions through future prediction. We train a future prediction model $f_\theta$, that maps the observation $o_t$ at time $t$, and a latent action $\hat{a}$ (from a set $\hat{\mathbf{A}}$ of discrete latent actions) to the observation $o_{t+1}$ at time $t + 1$, *i.e.* $f_\theta(o_t, \hat{a})$. $f$ is trained to minimize a loss $l$ between the prediction $f_\theta(o_t, \hat{a})$ and the ground truth observation $o_{t+1}$. $\hat{a}$ is treated as a latent variable during learning. Consequently, $f_\theta$ is trained using a form of expectation maximization (Bishop, 2006). Each training sample $(o_t, o_{t+1})$

---

[2] We use the terms state ($s_t$) and observation ($o_t$) interchangeably.

is assigned to the action that leads to the lowest loss under the current forward model. The function $f_\theta$ is optimized to minimize the loss under the current latent action assignment. More formally, the loss for each sample $(o_t, o_{t+1})$ is: $L(o_t, o_{t+1}) \coloneqq \min_{\hat{a} \in \hat{\mathbf{A}}} l\left(f_\theta(o_t, \hat{a}), o_{t+1}\right)$. We minimize $\sum_{(o_t, o_{t+1}) \in \mathbf{D}} L(o_t, o_{t+1})$ over the dataset to learn $f_\theta$.

Latent action $\hat{a}_t$ for observation pairs $(o_t, o_{t+1})$ are obtained from the learned function $f_\theta$ as: $\arg\min_{\hat{a} \in \hat{\mathbf{A}}} l\left(f_\theta(o_t, \hat{a}), o_{t+1}\right)$. Choice of the function $f_\theta$ and loss $l$ vary depending on the problem. We use L2 loss in the observation space (low-dimensional states, or images).

## 4.2 Q-LEARNING WITH LATENT ACTIONS

Latent actions mined from Section 4.1 allow us to complete the given $(o_t, o_{t+1}, r_t)$ tuples into $(o_t, \hat{a}_t, o_{t+1}, r_t)$ quadruples for use in Q-learning Watkins (1989). As our actions are discrete we can easily adopt any of the existing Q-learning methods for discrete action spaces (*e.g.* Mnih et al. (2013)). Though, we note that this Q-learning still needs to be done in an *offline* manner from pre-recorded state-only experience. While we adopt the most basic Q-learning in our experiments, more sophisticated versions that are designed for offline Q-learning (*e.g.* Kumar et al. (2020); Fujimoto et al. (2019)) can be directly adopted, and should improve performance further. Value functions are obtained from the Q-functions as $V(s) = \max_{\hat{a} \in \hat{\mathbf{A}}} Q(s, \hat{a})$.

## 4.3 BEHAVIORS FROM VALUE FUNCTIONS

Given a value function, our next goal is to derive behaviors from the learned value function. In general, this requires access to the transition function of the underlying MDP. Depending on what assumptions we make, this will be done in the following two ways.

**Densified Reinforcement Learning.** Learning a value function from state-only experience can be extremely valuable when a dense reward function for the underlying task is not readily available. In this case, using the learned value function can densify sparse reward functions, making previously intractable RL problems solvable. Specifically, we use the value function to create a *potential-based* shaping function $F(s, s') = V(s') - V(s)$, based on Ng et al. (1999), and construct an augmented reward function $r'(s, a, s') = r(s, a, s') + F(s, s')$. Our experiments show that using this densified reward function speeds up behavior acquisition.

**Domain Specific Low-level Controllers.** In more specific scenarios, it may be possible to employ hand designed low-level controllers in conjunction with a model that can predict the next state $s'$ on executing any of low-level controllers. In such a situation, behavior can directly be obtained by picking the low-level controller that conveys the agent to the state $s'$ that has the highest value under the learned $V(s)$. Such a technique was used by Chang et al. (2020). We show results in their setup.

## 5 EXPERIMENTS

We design experiments to assess the quality of value functions learned by LAQ from undirected state-only experience. We do this in two ways. First, we measure the extent to which value functions learned with LAQ without ground truth information agree with value functions learned with Q-learning with ground truth action information. This provides a direct quality measure and allows us to compare different ways of arriving at the value function: other methods in the literature (D3G (Edwards et al., 2020)), and simpler alternatives of arriving at latent actions. Our second evaluation measures the effectiveness of LAQ-learned value functions for deriving effective behavior in different settings: when using it as a dense reward, when using it to guide low-level controllers, and when transferring behavior across embodiments. Where possible, we compare to behavior cloning (BC) with *privileged* ground truth actions. BC with ground truth actions serves as an upper bound on the performance of state-only imitation learning methods (BCO from Torabi et al. (2018a), ILPO from Edwards et al. (2019), *etc.*) and allows us to indirectly compare with these methods.

**Test Environments.** Our experiments are conducted in five varied environments: the grid world environment from Section 3, the Atari game Freeway from Bellemare et al. (2013), 3D visual navigation in realistic environments from Chang et al. (2020); Savva et al. (2019), and two continuous control tasks from Fu et al. (2020)'s D4RL: Maze2D (2D continuous control navigation), and

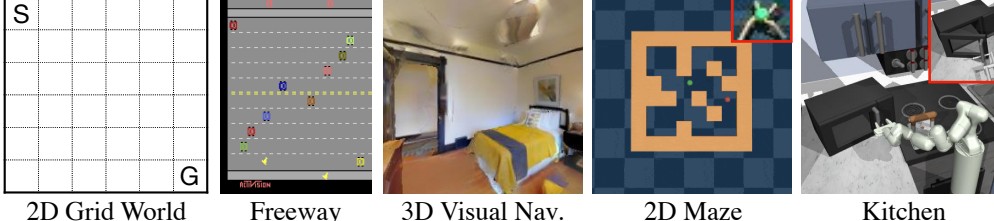

2D Grid World     Freeway     3D Visual Nav.     2D Maze     Kitchen

**Figure 3:** We experiment with five environments: 2D Grid World, Freeway (Atari), 3D Visual Navigation, Maze2D (2D Continuous Control), and FrankaKitchen. Top right corner of Maze2D and FrankaKitchen, shows the embodiments for cross-embodiment transfer (ant and hook, respectively).

**Table 1:** We report Spearman's correlation coefficients for value functions learned using various methods with DQN, against a value function learned offline using ground-truth actions (DQN for discrete action environments, and DDPG for continuous action environments). The *Ground Truth Actions* column shows Spearman's correlation coefficients between two different runs of offline learning with ground-truth actions. See Section 5.1. Details on model selection in Section A.11.

| Environment | D3G | Single Action | Clustering | Clustering (Diff) | Latent Actions | Ground Truth Actions |
|---|---|---|---|---|---|---|
| 2D Grid World | 0.959 | 0.093 | 0.430 | **1.000** | 0.985 | 1.000 |
| Freeway | – (image input) | 0.886 | 0.945 | 0.902 | **0.961** | 0.970 |
| 3D Visual Navigation | – (image input) | 0.641 | 0.722 | 0.827 | **0.927** | 0.991 |
| 2D Continuous Control | 0.673 | 0.673 | 0.613 | 0.490 | **0.844** | 0.851 |
| Kitchen Manipulation | 0.854 | 0.858 | 0.818 | 0.815 | **0.905** | 0.901 |

FrankaKitchen (dexterous manipulation in a kitchen). For Maze2D and FrankaKitchen environments, we also consider *embodiment transfer*, where we seek to learn policies for an ant and a hook respectively from the observation-only experience of a point mass and the Franka arm. Together, these environments test our approach on different factors that make policy learning hard: continuous control, high-dimensional observations and control, complex real world appearance, 3D geometric reasoning, and learning across embodiments. Environments are visualized in Figure 3, more details about the environments are provided in Section A.6.

**Experimental Setup** For each setting, we work with a pre-collected dataset of experience in the form of state, next state and reward triplets, $(o_t, o_{t+1}, r_t)$. We use our latent-variable forward model (Section 4.1) and label triplets with latent actions to obtain quadruples $(o_t, \hat{a}_t, o_{t+1}, r)$. We perform Q-learning on these quadruples to obtain value functions $V(s)$, which are used to acquire behaviors either through densified RL by interacting with the environment, or through use of domains-specific low-level controllers. We use the ACME codebase (Hoffman et al., 2020) for experiments.

**Latent Action Quality.** In line with the theory developed in Section 3, we want to establish how well our method learns a refinement of the underlying action space. To assess this, we study the *state-conditioned purity* of the partition induced by the learned latent actions (see definition in Section A.9). Overall, our method is effective at finding refinement of the original action space. It achieves higher state-conditioned purity than a single action and clustering. In high-dimensional image observation settings, it surpasses baselines by a wide margin. More details in Section A.9.

## 5.1 Quality of Learned Value Functions

We evaluate the quality of the value functions learned through LAQ. We use as reference the value function $V_{\text{gt-act}}$, obtained through *offline* Q-learning (DDPG for continuous action cases) with true ground truth actions *i.e.* $(o_t, a_t, o_{t+1}, r_t)$.[3] For downstream decision making, we only care about the relative ordering of state values. Thus, we measure the Spearman's rank correlation coefficient between the different value functions. Table 1 reports the Spearman's coefficients of value functions

---

[3] Offline DDPG in the FrankaKitchen environment was unstable. To obtain a reference value function, we manually define a value function based on the distance between the end-effector and the microwave handle (lower better), and the angle of the microwave door (higher better). We use this as the reference value function.

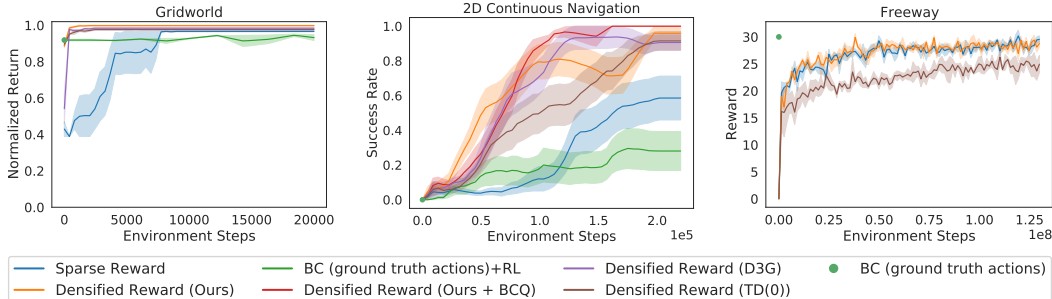

**Figure 4:** We show learning curves for acquiring behavior using learned value functions. We compare densified RL (Section 4.3) with sparse RL and BC/BC+RL. See Section 5.2 for more details. Results are averaged over 5 seeds and show ± standard error.

obtained using different action labels: single action, clustering, latent actions (ours), and ground truth actions. We also report Spearman's correlations of value functions produced using D3G (Edwards et al., 2020). In all settings we do Q-learning over the top 8 dominant actions, except for Freeway, where using the top three actions stabilized training. Our method out performs all baselines in settings with high-dimensional image observations (3D Visual Navigation, Freeway). In state based settings, where clustering state differences is a helpful inductive bias, method is still on-par with, or superior to clustering state differences and even D3G, which predicts state differences.

## 5.2 USING VALUE FUNCTIONS FOR DOWNSTREAM TASKS

Our next experiments test the utility of LAQ-learned value functions for acquiring goal-driven behavior. We first describe the 3 settings that we use to assess this, and then summarize our takeaways.

- **Using value functions as dense reward functions**. We combine sparse task reward with the learned value function as a potential function (Section 4.3). We scale up the sparse task rewards by a factor of 5 so that behavior is dominated by the task reward once policy starts solving the task. Figure 4 measures the learning sample efficiency. We compare to only using the sparse reward, behavior cloning (BC) with ground truth actions, and BC followed by spare reward RL.
- **Using value functions to learn behavior of an agent with a different embodiment.** Decoupling the learning of value function and the policy has the advantage that learned value functions can be used to improve learning across embodiment. We demonstrate this, we keep the same task, but change the embodiment of the agent in Maze2D and FrankaKitchen environments. Note that we do not assume access to ground truth actions in these experiments either. For Maze2D, the point-mass is replaced with a 8-DOF quadrupedal ant. For FrankaKitchen, the Franka arm is replaced with a position-controlled hook. We may need to define how we query the value function when the embodiment (and the underlying state space) changes. For the ant in Maze2D, the location (with 0 velocity) of the ant body is used to query the value function learned with the point-mass. For the hook in FrankaKitchen, the value function is able to transfer directly as both settings observe end-effector position and environment state. We report results in Figure 5.
- **Using value functions to guide low-level controllers.** Learned value functions also have the advantage that they can be used directly at test time to guide the behavior of low-level controllers. We do this experiment in context of 3D visual navigation in a scan of a real building and use the *branching environment* from Chang et al. (2020). We follow their setup and replace their value functions with ones learned using LAQ in their hierarchical policy, and compare the efficiency of behavior encoded by the different value functions.

**LAQ value functions speed up downstream learning.** Learning plots in Figure 4 show that LAQ-learned value functions speed up learning in the different settings over learning simply with sparse rewards (orange line *vs.* blue line). In all settings except Freeway, our method not only learns more quickly than sparse reward, but converges to a higher mean performance.

**LAQ discovers stronger behavior than imitation learning when faced with undirected experience.** An advantage of LAQ over other imitation-learning based methods such as BCO (Torabi et al., 2018a) and ILPO (Edwards et al., 2019) is LAQ's ability to learn from sub-optimal or undirected experience. To showcase this, we compare the performance of LAQ with behavior cloning (BC)

**Table 2:** We report Spearman's correlation coefficients for value functions learned using either DQN or BCQ, against a value function learned offline using BCQ with ground-truth actions. The *Ground Truth Actions* column shows Spearman's correlation coefficients between two different runs of offline learning with ground-truth actions. See Section 5.1.

| Environment | Single Action | Clustering | Clustering (Diff) | Latent Actions | Ground Truth Actions |
|---|---|---|---|---|---|
| 2D Continuous Control (DQN) | 0.664 | 0.431 | 0.312 | **0.807** | 0.765 |
| 2D Continuous Control (BCQ) | 0.710 | 0.876 | 0.719 | **0.927** | 0.990 |

**Figure 5:** Behavior acquisition across embodiments. Results averaged over 50 seeds and show ± standard error.

| | Interaction Samples | SPL |
|---|---|---|
| $V_{\text{one-act}}$ (Chang et al., 2020) | 0 | 0.53 |
| $V_{\text{cluster-act}}$ | 0 | 0.57 |
| $V_{\text{latent-act}}$ | 0 | 0.82 |
| $V_{\text{inverse-act}}$ (Chang et al., 2020) | 40K | 0.95 |

**Figure 6:** Visualization of trajectories and SPL numbers in the 3D visual navigation environment.

with ground truth actions. Since BCO and ILPO recover ground truth actions to perform behavior cloning (BC), BC with ground truth actions serves as an upper bound on the performance of all methods in this class. Learning plots in Figure 4 shows the effectiveness of LAQ over BC and BC followed by fine-tuning with sparse rewards for environments where the experience is undirected (Maze2D, and GridWorld). For Freeway, the experience is fairly goal-directed, thus BC already works well. A similar trend can be seen in the higher Spearman's coefficient for LAQ *vs.* $V_{\text{one-act}}$ in Table 1. LAQ discovers stronger behavior than imitation learning when faced with undirected data.

**LAQ is compatible with other advances in batch RL.** Although LAQ uses the most basic Q-Learning as our offline value learning method, it is compatible with recent more advanced offline RL value-learning methods (such as CQL (Kumar et al., 2020) and BCQ (Fujimoto et al., 2019)). We validate by simply swaping to using (discrete) BCQ with our latent actions. Figures. 4 and 5 show that LAQ with BCQ is the strongest method, outperforming ours with DQN, and D3G, on Maze2D and embodiment transfer environments. Analysis of Spearman's correlations in Table 2 shows the same trend as before with latent actions: better than single actions, and clustering variants. Note also that use of BCQ leads to value functions with better Spearman's correlations than DQN.

**LAQ value functions allow transfer across embodiments.** Figure 5 shows learning plots of agents trained with cross-embodiment value functions. LAQ-densified rewards functions, speed-up learning and consistently guide to higher reward solutions than sparse task rewards, or D3G.

**LAQ compares favorably to D3G.** We compare LAQ and D3G (a competing state-only method) in different ways. D3G relies on generating potential future states. This is particularly challenging for image observations, and D3G doesn't show results with image observations. In contrast, LAQ maps state transitions to discrete actions, and hence works with image observations as our experiments show. Even in scenarios with low-dimensional state inputs, LAQ learns better value functions than D3G, as evidences by Spearman's correlations in Table 1, and learning plots in Figure 4 and Figure 5.

**LAQ value functions can guide low-level controllers for zero-shot control:** We report the SPL for 3D navigation using value functions combined with low-level controllers in Figure 6. We report the efficiency of behavior induced by LAQ learned value functions as measured by the SPL metric from Anderson et al. (2018) (higher is better). The branching environment has two goal states, one optimal and one sub-optimal. The demonstrations there-in were specifically designed to emphasize the utility of knowing the intervening actions. Simple policy evaluation leads sub-optimal behavior (SPL of 0.53) and past work relied on using an inverse model to label actions (Chang et al., 2020) to derive better behavior. This inverse model itself required $40K$ interactions with the environment

for training, and boosted the SPL to 0.95. LAQ is able to navigate to the optimal goal (w/ SPL 0.82) but without the $40K$ online interaction samples necessary to acquire the inverse model. It also performs better than clustering transitions, doing which achieves an SPL of 0.57. The improvement is borne out in visualizations in Figure 6. LAQ correctly learns to go to the nearer goal, even when the underlying experience came from a policy that preferred the further away goal.

# 6 DISCUSSION

Our theoretical characterization and experiments in five representative environments showcase the possibility and potential of deriving goal-directed signal from undirected state-only experience. Here we discuss some scenarios which are fundamentally hard, and some avenues for future research.

**Non-deterministic MDPs.** Our theoretical result relies on a refinement where state-action transition probabilities are matched. However, the latent action mining procedure in LAQ results in deterministic actions. Thus, for non-deterministic MDPs, LAQ will be unable to achieve a strict refinement, leading to sub-optimal value functions. However, note that this limitation isn't specific to our method, but applies to *any* deterministic algorithm that seeks to learn from observation only data. We formalize this concept and provide a proof in Section A.3.

**Constraining evaluation of $V(s)$ to within its domain.** LAQ learns a value function $V(s)$ over the set of states that were available in the experience dataset, and as such its estimates are only accurate within this set. In situations where the experience dataset doesn't span the entire state space, states encountered at test time may fall out of the distribution used to train $V(s)$, leading to degenerate solutions. In Section A.12 we discuss a density model based solution we used for this problem, along with an alternate parameterization of value networks which helps avoid degenerate solutions.

**Offline RL Validation.** Validation (*e.g.* when to stop training) is a known issue in offline RL (Gulcehre et al., 2020). Like other offline RL methods, LAQ suffers from it too. LAQ's use of Q-learning makes it compatible to recent advances (Kumar et al., 2021) that tackle this validation problem.

# 7 RELATED WORK

Our work focuses on batch (or offline) RL with state-only data using a latent-variable future prediction model. We survey works on batch RL, state-only learning, and future prediction.

**Batch Reinforcement Learning.** As the field of reinforcement learning has matured, batch RL (Lange et al., 2012; Levine et al., 2020) has gained attention as a component of practical systems. A large body of work examines solutions the problem of extrapolation error in batch RL settings. Advances in these works are complementary to our approach, as substantiated by our experiments with BCQ. A more detailed discussion of batch RL methods can be found in Section A.1.

**State-only Learning.** Past works have explored approaches for dealing with the lack of actions in offline RL when given *goal-directed* or *undirected* state-only experience. Works in the former category rely on high quality behavior in the data, and suffer on sub-optimal data. Past work on state-only learning from undirected experience relies on either domain knowledge or state reconstruction and only show results with low dimensional states. See Section A.1 for continued discussion.

**Future Prediction Models.** Past work from Oh et al. (2015); Agrawal et al. (2016); Finn et al. (2016) (among many others) has focused on building action conditioned forward models in pixel and latent spaces. Yet other work in computer vision studies video prediction problems (Xue et al., 2016; Castrejon et al., 2019). Given the uncertainty in future prediction, these past works have pursued variational (or latent variable) approaches to make better predictions. Our latent variable future model is inspired from these works, but we explore its applications in a novel context.

**Latent MDP Learning** One way to interpret our method is that of learning an approximate MDP homomorphism (Taylor et al., 2008; Ravindran & Barto, 2004). Other works have explored learning latent homorphic MDPs. These methods tend to focus on learning equivalent latent state spaces (Li et al., 2006; Givan et al., 2003). Most similarly to our work (van der Pol et al., 2020) also learns a latent action space, but relies on ground truth action data to do so.

**Acknowledgement**: This paper is based on work supported by NSF under Grant #IIS-2007035.

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

# A  APPENDIX

## A.1  RELATED WORK CONTINUED

Discussion continued from Section 7:

**Batch Reinforcement Learning.** In recent times, Gulcehre et al. (2020) and Fu et al. (2020) propose datasets and experimental setups for studying offline RL problems. A large body of work examines solutions to the batch RL problem. Researchers have identified that *extrapolation error*, the phenomenon in which batch RL algorithms incorrectly estimate the value of states/actions not present in the training batch, is a major challenge, and have proposed methods to tackle it, *e.g.* BCQ (Fujimoto et al., 2019), BEAR (Kumar et al., 2019b), IRIS from (Mandlekar et al., 2020), and CQL (Kumar et al., 2020) among many others. In contrast to these model-free methods, Argenson & Dulac-Arnold (2021); Rajeswaran et al. (2019); Rafailov et al. (2020) learn a forward predictive model from the batch data and use it for model predictive control. These methods all approach the traditional batch RL problem, while we consider a different and harder setting in which the action labels are unavailable. Aforementioned advances in offline RL are complementary to our work. Offline value learning approaches (such as CQL and BCQ) can serve as a drop-in replacement for Q-learning in our pipeline and improve our results. In fact, our experiments with BCQ substantiate this.

**State-only Learning.** In line of work which studies learning form *goal-directed* state-only experience, researchers use imitation learning-based techniques (Radosavovic et al., 2020; Torabi et al., 2018a; Edwards et al., 2019; Kumar et al., 2019a), learn policies that match the distribution of visited states (Torabi et al., 2018b; 2019a;b), or use demonstrations to construct dense reward functions (Shao et al., 2020; Sermanet et al., 2017; Singh et al., 2019; Xie et al., 2018; Edwards & Isbell, 2019). These methods make strong assumptions about the quality and goal-directed nature of the experience data, and suffer in performance when faced with low-quality or undirected experience.

Instead of goal-directed experience our work tackles the problem of learning from undirected experience. Past work in this area employs Q-learning to learn optimal behavior from sub-optimal data (Chang et al., 2020; Song et al., 2020; Edwards et al., 2020). Chang et al. (2020) and Song et al. (2020) use domain specific insights. Edwards et al. (2020) rely on being able to generate the next state and only demonstrate results in environments with low-dimensional states. Instead, our work maps transition tuples to discrete latent actions and can thus easily work with high-dimensional observations such as RGB images.

A.2 PROOFS

*Proof.* **Proof for Lemma 3.2.** We will start by showing that for any policy $\pi_{\hat{M}}$ on $\hat{M}$, there exists a policy $\pi_M$ on $M$ such that $V_{\hat{M}}^{\pi_{\hat{M}}}(s) = V_M^{\pi_M}(s)$, for all $s$.

To do this we need to introduce the idea of *fundamental actions*, which are classes of actions which have the same state and reward transition distributions in a given state. If we have a fundamental action $b$, corresponding to some state and reward distributions, let $\alpha(b, s) \subseteq \mathbf{A}$ give the set of actions in $\mathbf{A}$ that have the matching state and reward transition distributions,

$$\underset{a \in \alpha(b,s)}{\forall} p(s', r|s, a) = p(s', r|s, \alpha(b, s)_1).$$

Similarly, let $\hat{\alpha}(b, s) \subseteq \hat{\mathbf{A}}$ give the set of actions in $\hat{\mathbf{A}}$ belonging to $b$ in state $s$. In any given state, there are at most $\min(|\mathbf{A}|, |\hat{\mathbf{A}}|)$ fundamental actions for $M$ and the union of all actions belonging to all fundamental actions gives the set of actions that make up the original action space. For our MDP let's denote $B(s)$ as the set of fundamental actions in the state $s$ for $M$, and $\hat{B}(s)$ as the set of fundamental actions in the state $s$ for $\hat{M}$. Let $\beta(s, a)$, and $\hat{\beta}(s, a)$ be functions which return the set of actions which correspond to the same fundamental action containing as $a$ in state $s$ for $M$, and $\hat{M}$ respectively. This means,

$$\bigcup_{b \in B(s)} \alpha(b, s) = A, \ \underset{b \in B(s)}{\forall} \ \underset{b' \neq b}{\forall} \ \alpha(b, s) \cap \alpha(b', s) = \emptyset, \text{ and } \underset{a \in A, s}{\forall} \exists_{b \in B(s)} | a \in \alpha(b, s). \quad (1)$$

With this notation out of the way we can construct a policy $\pi_{\hat{M}}$ from $\pi_M$, which achieves the same value in $\hat{M}$ as $\pi_M$ does in $M$. We do this by constructing $\pi_{\hat{M}}$ such that the probability distributions over each *fundamental action* is equivalent. We define this policy as

$$\pi_M(a|s) = \frac{1}{|\beta(s, a)|} \sum_{\hat{a} \in \hat{\beta}(s,a)} \pi_{\hat{M}}(\hat{a}|s).$$

Following Sutton & Barto (2018) we can define the value function for a given policy as

$$V_M^{\pi_M}(s) = \mathbb{E}_{\pi_M}[G_t | S_t = s],$$

$$V_M^{\pi_M}(s) = \sum_{a \in \mathbf{A}} \pi_M(a|s) \sum_{s',r} p(s', r|s, a)[r + \gamma V_M^{\pi_M}(s')].$$

From the properties in Eq. 1, we know a sum over fundamental actions will count each action exactly once, so we can write this sum as

$$V_M^{\pi_M}(s) = \sum_{b \in B(s)} \sum_{a \in \alpha(b,s)} \pi_M(a|s) \sum_{s',r} p(s', r|s, a)[r + \gamma V_M^{\pi_M}(s')]. \quad (2)$$

substituting in the constructed policy we have

$$V_M^{\pi_M}(s) = \sum_{b \in B(s)} \sum_{a \in \alpha(b,s)} \left[ \frac{1}{|\beta(s, a)|} \sum_{\hat{a} \in \hat{\beta}(s,a)} \pi_{\hat{M}}(\hat{a}|s) \right] \sum_{s',r} p(s', r|s, a)[r + \gamma V_M^{\pi_M}(s')].$$

since the third sum is over $\hat{\beta}(s, a)$ with $a \in \alpha(b, s)$, we can rewrite this sum as over $\hat{a} \in \hat{\alpha}(b, s)$, giving

$$V_M^{\pi_M}(s) = \sum_{b \in B(s)} \sum_{a \in \alpha(b,s)} \left[ \frac{1}{|\beta(s, a)|} \sum_{\hat{a} \in \hat{\alpha}(b,s)} \pi_{\hat{M}}(\hat{a}|s) \right] \sum_{s',r} p(s', r|s, a)[r + \gamma V_M^{\pi_M}(s')].$$

For the final sum, from the definition of fundamental actions, we know $p(s', r|s, a) = p(s', r|s, \alpha(b, s)_1) = \hat{p}(s', r|s, \hat{\alpha}(b, s)_1)$, so we can rewrite that term as

$$\sum_{s',r} \hat{p}(s', r|s, \hat{\alpha}(b, s)_1)[r + \gamma V_M^{\pi_M}(s')].$$

Crucially, in the second sum (over $a \in \alpha(b, s)$), $\beta(s, a)$ is the same for all $a \in \alpha(b, s)$, so the term in brackets can be treated as a constant. Similarly, since $|\beta(s, a)| = |\alpha(b, s)|$ we can simplify the second sum leaving

$$V_M^{\pi_M}(s) = \sum_{b \in B(s)} \sum_{\hat{a} \in \hat{\alpha}(b, s)} \pi_{\hat{M}}(\hat{a}|s) \sum_{s', r} \hat{p}(s', r | s, \hat{\alpha}(b, s)_1) \big[ r + \gamma V_M^{\pi_M}(s') \big].$$

This gives the definition of $V_{\hat{M}}^{\pi_{\hat{M}}}(s)$ using the same decomposition as equation Eq. 2, meaning

$$V_M^{\pi_M}(s) = V_{\hat{M}}^{\pi_{\hat{M}}}(s).$$

One can show the opposite direction, that a policy there exists a policy $\pi_{M'}$ for any policy $\pi_M$ such $V_M^{\pi_M}(s) = V_{\hat{M}}^{\pi_{\hat{M}}}(s)$, with a symmetric construction.

$\square$

Note that Lemma 3.2 holds true regardless of the stochasticity of the policy, as the constructed policy matches probability of the original policy for taking each fundamental action.

**Proof for Theorem 3.1.** Under the new MDP $\hat{M}$, Q-learning will learn the same optimal value function value function as learned under MDP $M$, *i.e.* $\forall_s V_{\hat{M}}^*(s) = V_M^*(s)$.

*Proof.* This follows from Lemma 3.2 by contradiction. Assume there is a state $s'$ where $V_{\hat{M}}^*(s) \neq V_M^*(s)$. Two cases must be considered. In the first case, $V_{\hat{M}}^*(s) > V_M^*(s)$. We will notate the optimal policy in $\hat{M}$ as $\tilde{\pi}_{\hat{M}}^*$ (*i.e.* $V_{\hat{M}}^* = V_{\hat{M}}^{\tilde{\pi}_{\hat{M}}^*}$) and the optimal policy in $M$ as $\pi_M^*$. We know there must exist a policy $\tilde{\pi}_M^*$ such that $V_M^{\tilde{\pi}_M^*}(s) = V_{\hat{M}}^{\tilde{\pi}_{\hat{M}}^*}(s)$ from Lemma 3.2. We arrive at a contradiction because $V_M^{\tilde{\pi}_M^*}(s) > V_M^{\pi_M^*}(s)$, so $\pi_M^*$ could not have been the optimal policy on $M$. The other direction follows a symmetric argument. If $V_{\hat{M}}^{\tilde{\pi}_{\hat{M}}^*}(s) < V_M^{\pi_M^*}(s)$, we know there must be a policy $\pi_{\hat{M}}^*$ in $\hat{M}$ such that $V_M^{\pi_M^*}(s) = V_{\hat{M}}^{\pi_{\hat{M}}^*}(s)$. This implies $V_{\hat{M}}^{\pi_{\hat{M}}^*}(s) > V_{\hat{M}}^{\tilde{\pi}_{\hat{M}}^*}(s)$, meaning that $\tilde{\pi}_{\hat{M}}^*$ could not have been the optimal policy. Since Q-learning is known to converge to the optimal value function Watkins (1989), Q-learning on $\hat{M}$ will converge to the original value function $V_M^*$. $\square$

A.3 STATE ONLY LEARNING FROM STOCHASTIC MDPS

Here we will briefly prove that problem of deducing the optimal value function of an MDP $M$, from some state-only experience dataset $D$, cannot be solved in general for non-deterministic MDPs. We notate $\mathbb{D}$ as the space of all *complete* state-only datasets, and $\mathbb{V}$ as the space of all value functions. By *complete* state-only dataset, we mean datasets in which all possible transition triples $(s, s', r)$ appear. Note that exactly one complete dataset exists for each MDP. We denote $D_M$ as the complete dataset corresponding to an MDM $M$.

In short, the idea is to construct a single state-only dataset which ambiguously could have been produced from two different MDPs which have different optimal value functions. Since no function deterministic function can model two outputs from the same input, no such function can exist.

**Theorem A.1.** *For any function $f : \mathbb{D} \to \mathbb{V}$ which maps from the set of of state-only datasets, $\mathbb{D}$, to a set of value functions. There exists some MDP $M$, with corresponding dataset $D_M$ such that $f(D) \neq V_M^*$.*

$$\nexists f \mid \forall_M f(D_M) = V_M^*.$$

*Proof.* We proceed by contradiction, assume any $f : \mathbb{D} \to \mathbb{V}$ such that

$$\forall_M f(D_M) = V_M^*.$$

If we can construct $M_1$ and $M_2$ such that $V_{M_1}^* \neq V_2^*$, and $D$ such that $D = D_{M_1} = D_{M_2}$, then $f$ cannot produce the correct value function for $f(D)$ in all cases, from the definition of a function. We will now show a very simple construction which exhibits this property. Consider an MDP $M_1$, with 3 states, $s_1, s_2, s_t$, and 2 actions $a_1, a_2$. $s_t$ is terminal, and gives reward 1, other states give reward 0. The transition dynamics of the two actions are given below.

**Table:** $a_1$ transitions for $M_1$          **Table:** $a_2$ transitions for $M_1$

|       | $s_1$ | $s_2$ | $s_t$ |
|-------|-------|-------|-------|
| $s_1$ | 0     | 1     | 0     |
| $s_2$ | 0     | 0     | 1     |

|       | $s_1$ | $s_2$ | $s_t$ |
|-------|-------|-------|-------|
| $s_1$ | 0     | 0     | 1     |
| $s_2$ | 1     | 0     | 0     |

For simplicity, assume $M_1$ has a discount factor of 0.9. This means $V^*(s_1) = V^*(s_2) = 1$.

We define $M_2$ as identical to $M_1$, only with stochastic transitions.

**Table:** $a_1$ transitions for $M_2$          **Table:** $a_2$ transitions for $M_2$

|       | $s_1$ | $s_2$ | $s_t$ |
|-------|-------|-------|-------|
| $s_1$ | 0     | 1     | 0     |
| $s_2$ | 0     | 0     | 1     |

|       | $s_1$ | $s_2$ | $s_t$ |
|-------|-------|-------|-------|
| $s_1$ |       | 0.9   | 0.1   |
| $s_2$ | 1     | 0     | 0     |

In this MDP, with a discount factor of 0.9, $V^*(s_2) = 1$, but $V^*(s_1) = 0.1 + 0.9 * 0.9 = 0.91$.

Since both $M_1$ and $M_2$ share the same states, actions, and possible transitions, $D_{M_1} = D_{M_2}$. So we have satisfied our condition that $D = D_{M_1} = D_{M_2}$, and $V_{M_1}^* \neq V^* M_2$. Thus, no $f$ can model the value functions for both MDPs in general. $\qquad\square$

### A.4 GRID WORLD BEHAVIOR *vs.* PURITY

Figure S7 plots the MSE in value function and the proportion of states with correct implied actions as a function of the noise in the action labels.

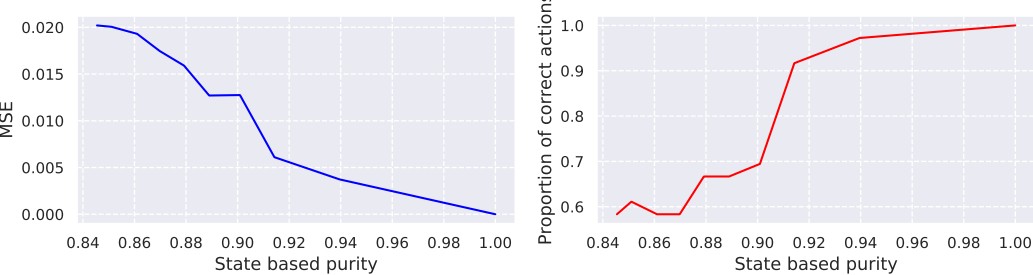

**Figure S7:** We plot correctness of value function estimate and behavior as a function of state-based purity of the intervening actions used for Q-learning. Left plot shows the mean squared error (lower is better) between the obtained value function and the optimal value function. Right plot shows fraction of states in which the learned value function induces the optimal action (higher is better). Both value function and behaviors become worse as state-based purity decreases.

## A.5 LAQ IN STOCHASTIC MDPS

We have conducted experiments on stochastic MDPs in the gridworld environment. Specifically, we do this by adding sticky actions, such that with a certain probability (called the stickiness parameter), the environment executes the previous action as opposed to the given action. We run LAQ with data from this stochastic environment and examine a) purity of latent actions, b) quality of learned value functions, and c) correctness of implied behavior.

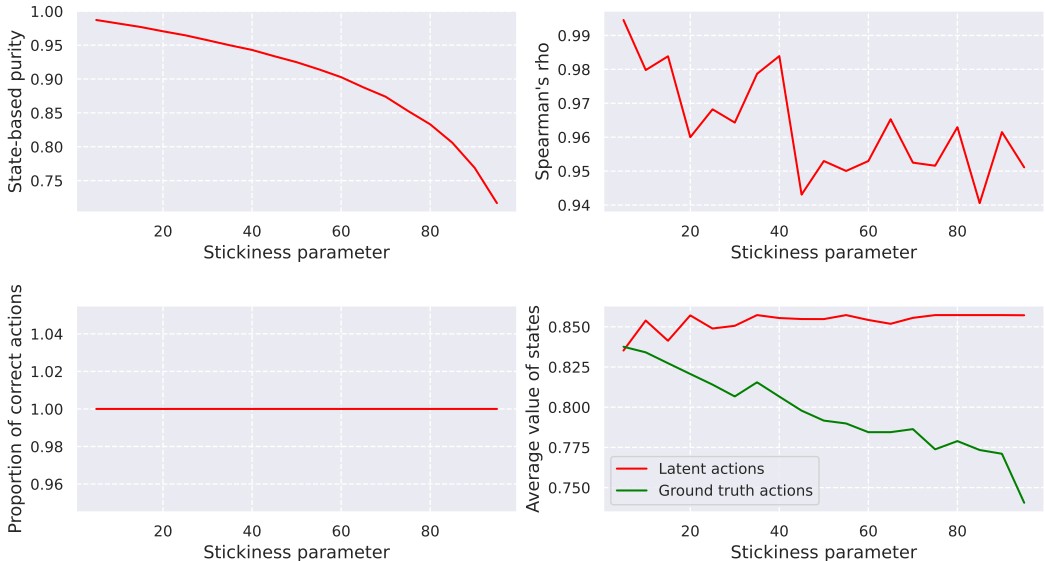

**Figure S8:** As a function of stochasticity in environment, we plot: state-based purity of the latent actions (top left), Spearman's rank correlation between LAQ learned value function and ground truth value function (top right), correctness of implied behavior by the LAQ learned value function (bottom right), and average state-value of LAQ learernd value function, and the ground truth value function (bottom left).

As expected, Figure S8 (left) shows that the state-based purity of the learned latent actions falls off as the stochasticity (stickiness) increases. However, these impure latent actions have little effect on the Spearman's rank correlation (top right). The Spearman's rank correlation does decrease but remains reasonably high even when the stickiness parameter is set to 95%. While the ordering of the values of the different states doesn't change by much, we note increasing amount of over-estimation in the values (bottom right). This is expected as with increasing stochasticity, there is an increasingly large gap between what LAQ thinks it can control, and what it can actually control. However, the behavior implied by LAQ is still always correct (bottom left), as the over-estimation is uniform over all states. Thus, it is possible to get good performance from LAQ in some stochastic environments. However, it is possible to also construct simple scenarios where LAQ, and for that matter any deterministic algorithm, will suffer as we describe in Section A.3.

## A.6 ENVIRONMENT DETAILS

**2D Grid World.** We use the environment and data from Section 3.2. We use the $(x, y)$ coordinates as our state for this environment. We use a multi-layer perceptron for $f_\theta$ and L2 loss for $l$.

**2D Continuous Control Navigation.** We use the Maze2D data from D4RL Fu et al. (2020). Observation space here is $(x, y, v_x, v_y)$ *i.e.* location and velocity of the agent, and action space is force along $x$ and $y$ directions. We use a frame skip of 3. We use a multi-layer perceptron for $f_\theta$ and L2 loss for $l$. In the embodiment transfer variant, we swap the agent for a 4-legged ant, also from D4RL. The ant embodiment is far more challenging to control, with an 8-d action space and 29-d observation space.

**Atari Game.** We work with Freeway. We generated our own data using the protocol described in Agarwal et al. (2020). We turned off sticky actions and store frames both at the default resolution $(84 \times 84)$ and full resolution $(224 \times 224)$. Other settings are same as is typical: stack last 4 frames to represent the observations, and use frame-skip of 4. We use a convolutional encoder-decoder model for $f_\theta$ and predict raw future observations. We use L2 distance in the pixel space as our loss function $l$.

**3D Visual Navigation.** We work with the branching environment from Chang et al. (2020) in the AI Habitat Simulator Savva et al. (2019), and use the provided dataset of first-person trajectories as our pre-recorded experience dataset. Following Chang et al. (2020), we employ low-level controllers for deriving behaviors from the learned value functions. We use a convolutional encoder-decoder for $f_\theta$, and L2 loss in pixel space as $l$.

**Kitchen Manipulation.** In this task a 9-DOF Franka arm can interact with several different objects physically simulated kitchen. The observations are 24-d, containing the end-effector position and state of various objects in the environment (microwave door angle, kettle position, etc.). The action space represents the 9-d joint velocity. In cross-embodiment experiments we replace the Franka arm with a hook. The observation space remains the same while the action space becomes 3-d position control.

For experiments in kitchen manipulation, we utilize the GMM based solution described in Section A.12. For all environments, we throw out the provided action labels when learning our models.

## A.7 DATA COLLECTION

**2D Grid World.** The setting for this experiment is a $6 \times 6$ grid with 8 actions, corresponding to moving in the 4 cardinal directions and 4 diagonal directions. The agent starts in the top left (0,0), and gets reward 0 everywhere, except for in the bottom right, (5,5). Reaching the bottom right gives the agent reward 1 and terminates the episode. In the starting square (0,0), the agent has probability 0.5 of moving right, and probability 0.5 of moving down. When the agent is on the top or bottom edge of the maze (row = 0 or row = 5) it moves right with probability 0.9, and takes a random action with probability 0.1. When it is on the left or right edge of the grid (column = 0 or column = 5), it moves down with probability 0.9 and takes a random action with probability 0.1. Otherwise, it is in the interior of the grid, where it takes an action away from the goal (randomly chosen from: up, left, or up-left) with probability 0.9, and one of the remaining actions otherwise (randomly chosen from: down, right, down-right, up-right, down-left). This policy is rolled out for 20,000 policies to generate the data for methods described in main paper Section 3.2 and Figure 1.

**Atari Game.** Data for the Freeway environment had to be additionally collected, as native high resolution resolution images of the episodes from Agarwal et al. (2020) are not readily available. For this, we re-generated the data using the protocal described in Agarwal et al. (2020), without sticky actions. Then, the high resolution data was collected by taking the action sequences from the re-generated dataset and executing them in an emulator, while storing the native resolution images. Because we re-generated the data without sticky actions, the high resolution episodes are perfect recreations of the low resolution episodes.

**Visual Navigation.** The data used for the visual navigation experiment was generated using the same protocol as Chang et al. (2020). Agents are tasked to reach one of two goals ($G_{near}$ and $G_{far}$) in a visually realistic simulator Savva et al. (2019). Navigating to $G_{near}$ is the optimal path as it is nearer to the agent's starting location.

The dataset contains 3 types of trajectories, $T_1$, $T_2$, and $T_3$, representing 50%, 49.5%, and 0.5% of the trajectories in the dataset respectively . $T_1$ takes a sub-optimal path to $G_{near}$, $T_3$ takes the optimal path to $G_{near}$, and $T_2$ navigates to $G_{far}$.

**2D Continuous Control (Maze2D)** We use the *Maze2D* dataset from D4RL as is.

**Kitchen Manipulation.** The data used for the Kitchen Manipulation environment comes from the *partial* version of the D4RL FrankaKitchen dataset. To facilitate cross-embodiment transfer we convert the state representation to contain end-effector location instead of joint angles.

## A.8   ALGORITHM

---

**Algorithm 1** LAQ

---

1: Given dataset $D$ of $(o_t, o_{t+1}, r)$ triples
2: **for** each epoch **do**                                        ▷ Latent Action Mining
3:     **for** sampled batch $(o_t, o_{t+1}, r) \sim D$ **do**
4:         $L(o_t, o_{t+1}) \leftarrow \min_{\hat{a} \in \hat{\boldsymbol{A}}} l(f_\theta(s, \hat{a}), s')$
5:         $\theta \leftarrow \theta - \alpha \nabla_\theta L(o_t, o_{t+1})$          ▷ Section 4.1
6:     **end for**
7: **end for**
8: **let** $\hat{g}(o_t, o_{t+1}) = \arg\min_{\hat{a} \in \hat{\boldsymbol{A}}} l(f_\theta(o_t, \hat{a}), o_{t+1})$
9: $\hat{D} = \{(o_t, o_{t+1}, r, \hat{g}(o_t, o_{t+1})) \mid (o_t, o_{t+1}, r) \in D\}$          ▷ Latent Action Labeling
10: **for** each epoch **do**
11:     **for** sampled batch $(o_t, o_{t+1}, r, \hat{a}) \sim \hat{D}$ **do**          ▷ Q-learning with Latent Actions
12:         Q-learning update to learn $Q(s, \hat{a})$
13:     **end for**
14: **end for**
15: $V(s) = \max_{\hat{a} \in \hat{\boldsymbol{A}}} Q(s, \hat{a})$

---

A.9 LATENT ACTION QUALITY

We first measure the effectiveness of our latent action mining process by judging the extent to which the induced latent actions are a refinement of the original actions. We measure this using the *state-conditioned purity* of the partition induced by the learned latent actions.

In a given state, for any latent action $\hat{a}$, there must be some ground truth action which most frequently mapped to $\hat{a}$. We define the purity of $\hat{a}$ as the proportion of the most frequent action among all actions mapped to $\hat{a}$. For example, in a given state $s$ if a set of actions $[0, 0, 0, 1, 2]$, were mapped to latent action $\hat{a}$, then $0$ is the most frequent action mapped to $\hat{a}$ and thus the purity of $\hat{a}$ would be $0.6$. For a given state, the purity of an entire set of latent actions is the weighted mean of purity of individual latent actions. Overall purity is the average of all state wise purities weighted by how often the states appear in the dataset.

We extend this definition of state-conditioned purity to continuous or high-dimensional states by measuring the validation accuracy of a function $g$ that is trained to map the high-dimensional state (or observation) $s$, and the associated latent action $\hat{a}$ to the actual ground truth action $a$. Training such a function induces an implicit partition of the state space. Learning to predict the ground truth action from the latent action $\hat{a}$ within this induced partition estimates the most frequent ground truth action, and accuracy measures its proportion, *i.e.* purity. This exact procedure reduces to the above definition for discrete state spaces, but also handles continuous and high-dimensional states well. For continuous action spaces, we measure the mean squared error in action prediction instead of classification accuracy.

Table S3 reports the purity (and MSE for continuous action environment) obtained by our proposed future prediction method. For reference, we also report the purity obtained when using *single action* that maps all samples into a single cluster, and 2 *clustering* methods that cluster either concatenation of the two observations *i.e.* $[o_{t+1}; o_t]$, or the difference between the two observations *i.e.* $[o_{t+1} - o_t]$. We use 8 latent actions for all environments except the FrankaKitchen environment for which we use 64 because of its richer underlying action space.

In general, our forward models are effective at generating a latent action space that is a state-conditioned refinement of the original action space. This is indicated by the improvement in state-conditioned purity values over using a single action or naive clustering. For the 2D Grid World and 2D Continuous Navigation, clustering in the correct space (state difference *vs.* state concatenation) works well as expected. But our future prediction model, which directly predicts $s_{t+1}$ and doesn't use any domain specific choices, is able to outperform the corresponding clustering method. We also observe large improvements over all baselines for the challenging case of environments with high-dimensional state representations: Freeway and 3D Visual Navigation.

**Table S3:** We report the state-conditioned action purity (higher is better, MSE for continuous action case where lower is better), of latent actions for different approaches: single action, clustering concatenated observations, clustering difference in observations, and the proposed future prediction models from Section 4.1. We note the utility of the future prediction model for the challenging case of Freeway and 3D Visual Navigation environments. See Section A.9 for a full discussion.

| Environment | Observation Space | Action Space | Purity Metric | Single Action | Clustering $[o_t, o_{t+1}]$ | Clustering $[o_{t+1} - o_t]$ | Future Prediction |
|---|---|---|---|---|---|---|---|
| 2D Grid World | $xy$ location | Discrete, 8 | Purity ($\uparrow$) | 0.827 | 0.851 | **1.000** | 0.998 |
| Freeway | $210 \times 160$ image | Discrete, 3 | Purity ($\uparrow$) | 0.753 | 0.778 | 0.773 | **0.907** |
| 3D Visual Navigation (Branching) | $224 \times 224$ image | Discrete, 3 | Purity ($\uparrow$) | 0.783 | 0.839 | 0.859 | **0.928** |
| 2D Continuous Control | $xy$ loc. & vel | Continuous, 2 | MSE ($\downarrow$) | 2.207 | 2.188 | **0.325** | 0.905 |
| Kitchen Manipulation | 24-d State | Continuous, 8 | MSE ($\downarrow$) | 0.015 | 0.015 | 0.015 | **0.014** |

## A.10 ANALYSIS OF LEARNED LATENT ACTIONS

We analyze the latent actions learned for the Freeway environment.

We visualize our future prediction model's predictions for the Freeway environment in Figure S9. In line with our expectations, the one action future prediction model and the latent action future prediction model are both able to reconstruct the background and the vehicles perfectly. At the same time, the one action future prediction model fails to reconstruct the agent accurately, whereas the latent action future prediction model is able to reconstruct the agent almost perfectly. This provides evidence for the effectiveness of our proposed latent action mining approach at discovering pure action groundings.

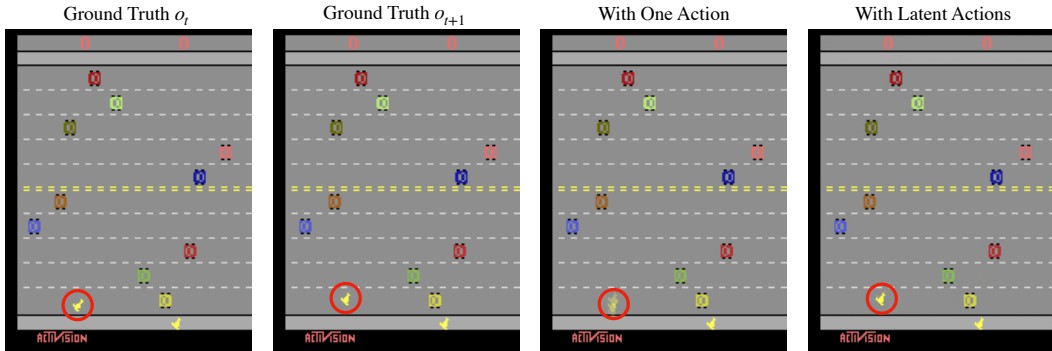

**Figure S9:** We visualize the Freeway future prediction models' reconstructions for $o_{t+1}$. From left to right, the ground truth $o_t$, the ground truth $o_{t+1}$, the reconstruction for $o_{t+1}$ by the future prediction model with one action, the reconstruction for $o_{t+1}$ by the future prediction model with latent actions. The agent is circled in each image.

Furthermore, we visualize the learned latent actions for Freeway in Figure S10. In Freeway, the agent can only move along the y-axis, and consequently, the environment action space only has three actions: move up, move down, and no-op. This means that the agent's y-displacement between the current frame and the previous frame directly corresponds to the ground truth action taken. In this visualization, we visualize the the chosen latent action (by color: blue, orange, or green) as a function of the agent's y-position in the current frame (x-axis) and the y-displacement relative to the previous frame (y-axis). Note that as mentioned in the paper, we learn the value functions over the top three most dominant actions to stabilize training; for this reason, the visualization only consists of three latent actions.

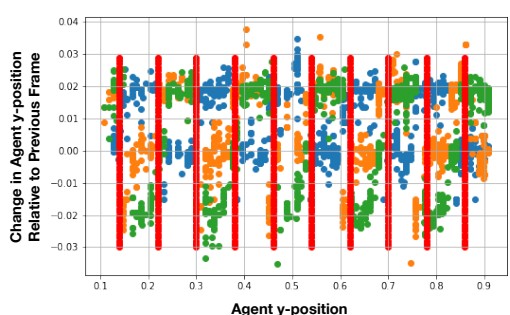

**Figure S10:** Visualization of the latent actions learned in Freeway. X-axis is the agent's y-position, y-axis is the displacement of the agent (which is indicative of ground truth action).

Within each vertical region marked by the red bars, we see that the latent actions are split into distinct clusters based on the agent's y-displacement. Because the agent's y-displacement directly corresponds to the ground truth action taken, this indicates that given the agent's y-position, the learned latent actions encode information about the ground truth action. Hence, we qualitatively confirm that the state-based purity of the latent actions is high.

## A.11 VALUE FUNCTION DETAILS

**Value Function Model Selection** The numbers reported in Table 1 are the 95th percentile Spearman's correlation coefficients over the course of training. If training is stable and converges, this corresponds to taking the final value, and in the case that training is not stable and diverges, this acts as a form of early stopping. We take the 95th percentile as opposed to the maximum to eliminate outliers.

Below we present visualizations of value functions learned from LAQ.

**Freeway.** We visualize the value functions learned using our latent actions for Freeway. Figure S11 plots the values over the course of an episode. In Freeway, the agent has to move vertically up and down the screen to cross a busy freeway, receiving reward when it successfully gets to the other side. In a single episode, the agent can cross the freeway multiple times; each time the agent makes it to the other side, the agent's location is reset to the original starting location, allowing the agent to attempt to cross the freeway once again. For this reason, we see the value increase as the agent gets closer to the other side of the road, and then drop as soon as its position resets to the starting location. As evident, the peaks of the learned value function correspond highly to the environment reward.

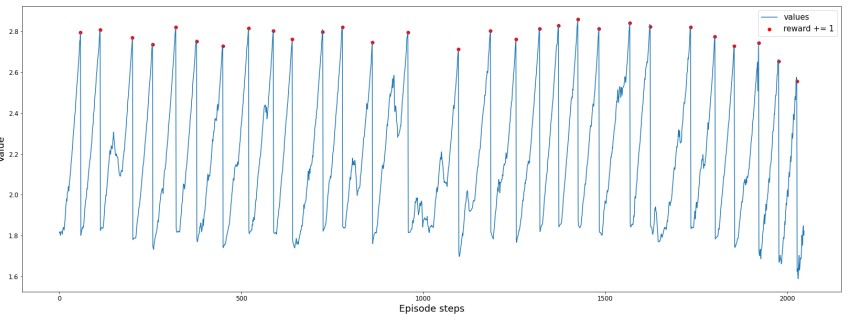

**Figure S11:** We visualize the value predicted by our learned value function over the course of one episode of Freeway. The red points correspond to when the agent receives a reward from the environment.

**3D Visual Navigation.** Figure S12 visualizes the learned value map in the 3D Visual Navigation branching environment from Chang et al. (2020). As depicted in Figure 6, there are two goals: $G_{near}$ and $G_{far}$. The figure on the right illustrates that the value function learned by utilizing our latent actions correctly assigns high value to the regions surrounding the two goal locations, and low value elsewhere. Additionally, we learn the value functions with DQN using a dataset obtained by using a sub-optimal policy which prefers to go to the goal further away rather than the goal close by. Despite this sub-optimality, the learned value function correctly assigns a higher value to the nearby goal than the goal which is further away.

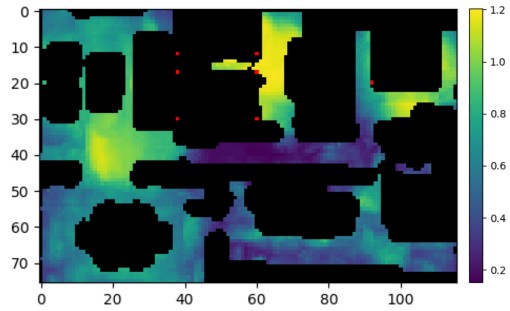

**Figure S12:** Visualization of the learned value function for 3D Visual Navigation.

**2D Continuous Control.** Figure S13 shows a visualization of the value function learned using latent actions for the 2D Continuous Control environment. While the observation space of this environment is $(x, y, v_x, v_y)$ *i.e.* location and velocity of the agent, we produce this visualization over just the location of the agent. Based on just the location of the agent, the optimal value function would be a monotonically decreasing function as the distance from the goal location increases. In this particular environment, the goal of the agent is to get to the top-right corner of the maze. The visualization shows that the learned value function produces high values around this goal region at the top-right corner of the maze, and gradually lower values the farther away you go. This figure visually is in line with our quantitative results in Table 1 which show that the value function learned using latent actions in the 2D Continuous Control environment highly correlates to that learned using ground truth actions.

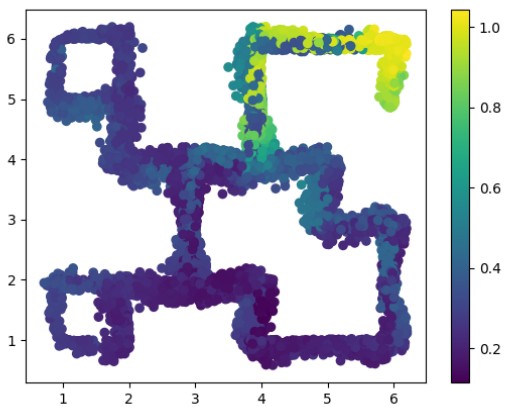

**Figure S13:** Visualization of the learned value function for 2D Continuous Control.

**Kitchen Manipulation.** We visualize the values as a function of both the angle that the microwave door makes with respect to its starting state as well as the distance between the end-effector and the microwave door handle in Figure S14. The task here is to open the microwave door: the agent receives a binary reward when the angle that the microwave door makes with respect to its starting state (i.e. closed microwave door) is above a threshold (approx. 0.7 radians), and zero otherwise. As expected, we see that the predicted value of a state increases as the angle of the microwave door increases. The end-effector position does not start out at the microwave door handle, but rather is initialized a fixed distance away from the handle. As a result, the agent must first move the end-effector to the microwave door handle, before interacting with the door handle to open it. In addition to this, we hypothesize that as the distance between the end-effector and the microwave door handle decreases, the predicted value should increase. While not as obvious as the relation between the microwave door angle and the value, we see some indication that as the distance between the end-effector and the microwave door handle decreases, the value increases.

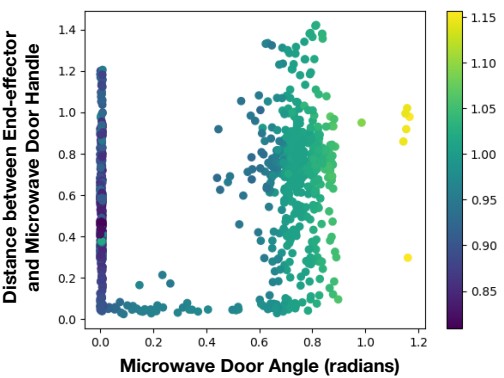

**Figure S14:** Visualization of the learned value function in Kitchen Manipulation. The x-axis is the microwave angle in radians, the y-axis is the distance between the end-effector and the microwave door handle, with the colors corresponding to the magnitude of the values.

## A.12 DEALING WITH EXTRAPOLATION ERROR

Given the reward structure utilized in densified reinforcement learning (see Section 4.3), if there exist states which produce spuriously high values of $V(s)$, the optimal policy in the new MDP may erroneously seek these states. Here we present two ways to combat this phenomenon:
**1)** Fit a density model to constrain high reward regions to the distribution used to train $V(s)$
**2)** Re-parameterize the value function such that out of distribution states are likely to have low value.

**Density Model:** We add an additional component to the reward computation to incentivize behavior to remain within the region of the state space covered by the demonstrations (where the learned value function is in-distribution). For the kitchen manipulation environment we build a density model over the end effector position using a 2 component Gaussian mixture model. If a state $s$ has a less than 1% probability according to the GMM density model, then we assign $V(s) = 0$, for the update described in Section 5.2. We apply this shaping to all methods in Figure 4 for the kitchen environment. We give sparse reward the same benefit of this shaping by giving reward $-1$ outside of the GMM distribution, and $0$ inside. Sparse task reward remains the same.

**Value function re-parameterization** For sparse reward tasks, we can parameterize the final value prediction as $1 - \|\phi(s)\|_2$. Where $\phi$ is a neural network based featurization of the state. This ensures high values are only predicted when the featurization of the state is close to the zero vector. States which are out of distribution may induce random features, but these will result in lower predicted values.

Results on the kitchen manipulation environment are presented in Figure S15. We can see that all methods solve the task poorly in this environment when value functions are unconstrained. Using a density based model (GMM) or the "1-norm" parameterization of the value function allow the task to be solved consistently.

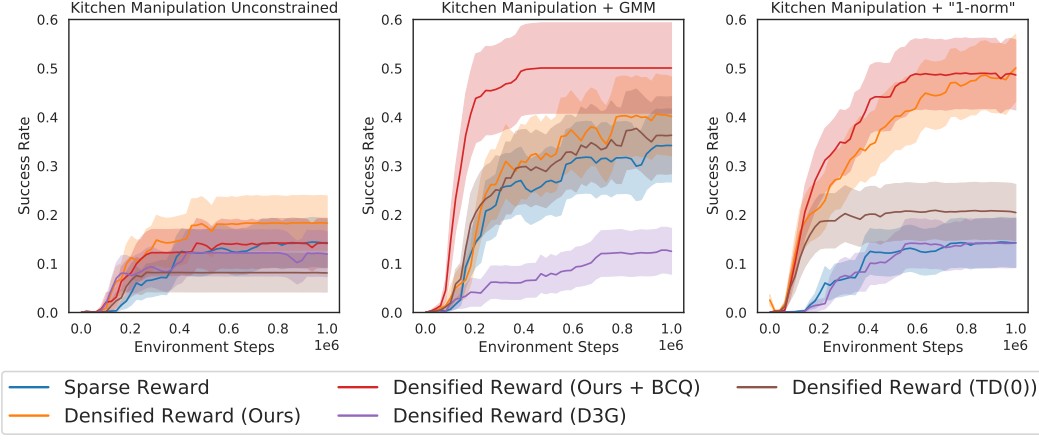

**Figure S15:** We show learning curves for acquiring behavior using different methods for dealing with out of distribution samples in the kitchen manipulation environment: unconstrained (left), density model based reward shaping (center), and "1-norm" parameterization of the value function (right). Results are averaged over 5 seeds and show 95% confidence intervals. See Section A.12 for details.

