# OpenReview forum: "Learning Value Functions from Undirected State-only Experience"
_ICLR.cc/2022/Conference — ICLR 2022 Poster_

### Official Review · Reviewer_kJe7 · 2021-10-29

**Correctness:** 3
**Technical Novelty And Significance:** 2
**Empirical Novelty And Significance:** 2
**Recommendation:** 5
**Confidence:** 2

**Main Review:**

Some comments on motivation, related work, structure, etc.:

Motivation: I think the motivation is somewhat overstating the difficulty of the problem. Isn't it obvious that state-only often helps? I think there are enough common model examples where the action is even fully identifiable from just the current and next state (just think of a double integrator dynamics).

I'm OK wuth havung the extensuve related work section at the end, but at least having a rough idea of existing work in this direction up front would be important, this is missing.

I think the "refinement" is closely related to MDP homomorphisms and commutation of operators etc., but this seems entirely missing in the related work section.

Some comments on definitions, problem formulation etc.:

I'm a bit confused. There seem to be two issues: unobserved action, but "same embodiment", or unobserved actions and even allowing for different embodiments. It may well be that what is correctly addressed is the second case, but it would be helpful to at least discern the two things in the discussion.

The defi of "refinement" seems central so why put this in inline text, not as a Definition? And the inline formulation itself is very hard to parse, I think the second part of the sentence is somewhat incomplete. When continuing reading, it seems like "refinement" is formalized

Where is the value function V defined? This should be done explicitly in sec2.

Regarding Theorem 1:

It would also help to motivate this better -- it seems that this theorem should motivate the subsequent algorithm. Personally, I would have expected a more principled result as the basic theorem in the sense of "to what extent is the true MDP recoverable from just the given data. I think this is somehow implicated by theorem 1, but it may be worth making this more explicit and systematic.

**Summary Of The Paper:**

They address undirected (i.e., the data was not necessarily generated by an optimal agent) offline RL in the case that not actions, but only state transitions plus rewards are observed. They present a theoretical result that motivates their subsequent method for finding the true Q function from just the given data. In experiments, they evaluate the method in common toy environments like gridworld.

**Summary Of The Review:**

While the problem setting, i.e., state-only offline RL, is realistic and practically relevant (cheaper that classic RL), and the paper does contain some interesting ideas in terms of theory and algorithm, the paper is too premature. The writing sometimes lacks motivations and connections, and the theory, I think, can actually be extended (be made more systematic) already within this paper, this would be a deeper contribution.

---

> ### Author Response · Authors · 2021-11-20
> **Reply to reviewer kJe7**
>
> Thank you for your thoughtful comments and feedback. We will respond to your specific concerns below, but before that we wanted to note that our experiments, presented in Section 5, were done in 5 different environments and not just toy environments like grid world (as noted in your summary of our paper). These include Gridworld, 2D continuous navigation (with point mass and quadrupedal ant), Freeway (Atari), Kitchen manipulation (with Franka hand and hook), 3D Visual Navigation. Together, these environments test our approach on factors that make policy learning hard: continuous control, high-dimensional observations and control, complex real world appearance, 3D geometric reasoning, and learning across embodiments. We regard empirical results on these challenging environments as important contributions alongside our theoretical results which motivated the design of our LAQ method in Section 4.
>
> **Motivation is overstating the difficulty of the problem.** You are right that some of our wording was too strong. We have fixed this in the revision in Section 1. We would like to note that while it is true that the action is fully identifiable from just the current and next state in some environments, this is not true for many realistic settings. In particular, it is difficult to extract this kind of information from visually rich environments with high-dimensional state spaces. We present a methodology to do this in a completely self-supervised manner, and show that our approach achieves strong experimental results on a number of challenging environments. Furthermore, this is a non-trivial problem setting which has been tackled in the past by numerous papers (e.g. Radosavovic et al., 2020; Torabi et al., 2018a; Edwards et al., 2019; Kumar et al., 2019a; Torabi et al., 2018b; Torabi et al., 2019a; Torabi et al., 2019b; Shao et al., 2020; Sermanet et al., 2017; Singh et al., 2019; Xie et al., 2018; Edwards & Isbell, 2019).
>
> **Briefly mention related work at the beginning.** We appreciate your feedback that our introduction should include further discussion of related work. In our updated draft we have included additional language to this effect.
>
> **MDP homomorphisms and commutation of operators in related work.** Thank you for bringing this to our attention. We have added relevant citations to the text. We would be happy to include any specific references the reviewer would like to recommend.
>
> We respond to the specific comments on definitions, problem formulation etc. below:
>
> **Discerning between “same embodiment, unobserved actions” and “different embodiment, unobserved actions”.** Yes, all of our experiments are in the “unobserved action, but same embodiment” setting, except for the transfer of embodiment experiments, which are in the “unobserved action, and different embodiment” setting. We have made this distinction more clear in the revised version in the main text in Section 5.2.
>
> **Definition of refinement.** Yes, the revised version has added refinement as a Definition in Section 3. We have also rephrased the definition for clarity.
>
> **Definition of value function.** We have added the definition of a value function to Section 2. Please see the revised version of the paper. If you were asking about how we obtain value functions from LAQ, we describe that in Section 4.2.
>
> **Extension for Theorem 1.** This is an excellent suggestion. We view Theorem 1 to be a non-trivial contribution which aids in motivating the methodology. It is important to note that our aim is to recover optimal value functions that lead to optimal behavior, and not necessarily to recover the true underlying MDP. We empirically demonstrate that good value functions and high reward behavior can be derived from our approach, but a theoretical characterization of "to what extent is the true MDP recoverable from just the given data” is currently beyond the scope of this paper.

---

### Official Review · Reviewer_Fa3L · 2021-11-02

**Correctness:** 3
**Technical Novelty And Significance:** 3
**Empirical Novelty And Significance:** 4
**Recommendation:** 8
**Confidence:** 3

**Main Review:**

I found the problem setting to be both technically and conceptually interesting. In addition to its motivation, I found the paper well written and easy to read overall. Ideas are introduced carefully (with the small exception of some details in Lemma 3.2), and I believe this paper will be broadly accessible to the RL community. The two presented theoretical results are quite compelling in their own right, if quite natural in hindsight. Some of the claims made (the paragraph headers in the experimental section) seemed too strong given the data. For instance, in the paragraph "LAQ value functions allow transfer across embodiments", it is stated that "LAQ-densified rewards functions, speed-up learning and consistently guide to higher reward solutions than sparse task rewards, or D3G". Looking at Figure 5, the confidence intervals are large enough that I believe it would be difficult to definitively conclude that orange strictly outperforms purple. I don't believe the overlapping intervals detracts from the paper, but I do believe that the paper would be stronger if such claims were of appropriate strength given the data.

I have no major concerns about the paper, and only one primary question. I include several writing suggestions below. I am not intimately familiar with state-only learning in RL, so unless this idea has already been examined elsewhere in the literature, I believe this paper is ready for publication (pending a few changes suggested below, and any other reservations that other reviewers might have).

**Major Comments/Questions**.
- [Q.1] In the discrete action case, it was unclear to me whether $n$ is known. That is, the number of actions in the true environmental model. This seems important for determining how to perform the latent action inference.
- [C.1] Including the pseudocode for LAQ would benefit the paper a lot.

**Minor Questions**
- Does Lemma 3.2 quantify over stochastic policies, or only deterministic ones? Ah, in looking at the proof in the appendix, it seems that these are necessarily _stochastic policies_. It would be helpful to state this explicitly in the Lemma in the main text.

**Writing Suggestions**
- An extremely minor point, but I consider $(s,a,r,s')$ to be such a canonical ordering that the initial $(s_t, a_t, s_{t+1}, r_t)$ read oddly to me.
- "bellman equation" --> "Bellman equation"
- It would be helpful to define "refinement" formally. So, use a definition block and state it precisely.
- I found the idea of a "fundamental action" to be quite sharp and added a lot of clarity to thinking about the proof. It is not strictly necessary, but if you found a way to make space, it might be nice to include a brief proof sketch for Lemma 3.2 in the main paper that highlights this idea.
- "induce same value in every state" --> "induce the same value in every state
- "in 2 ways." --> "in two ways."
- In the "Experimental Setup" paragraph, we move to using $o$ in place of $s$. I did not quite understand this move, and would suggest sticking with $s$.
- "in 5 representative environments" --> "in five representative environments"

(Appendix)
- "at at most" --> "at most"
- "be a functions which" --> "be functions which"
- I did not quite understand this: "we know a sum over action in each fundamental action", but perhaps changing to the following will help: "we know a sum over fundamental actions"
- A minor point, but many equations in the appendix do not have punctuation. I would recommend adding commas and periods to equations where appropriate.
- In Proof of Theorem 3.1: "First, is the case where" --> "In the first case,"
- This is just personal taste, but I prefer $\neg \exists$ notation to $\not \exists$ (I am referring to Theorem A.1)

**Summary Of The Paper:**

This paper studies the offline reinforcement learning problem based only on trajectories without actions. That is, given a set of experiences of the form $(s_t, r_t, s_{t+1})$, how might a learning algorithm proceed? The paper takes care in motivating the problem, both from a conceptual perspective (we might not know the action interface of an agent), and technically (there is good reason we might have access to such trajectories of experience, and learning seems feasible). Given this setting, the paper's main focus is twofold: First, to establish a simple variant of Q-learning that is well-behaved under this learning problem. This motivates Theorem 3.1/Lemma 3.2, which in turn inform the design of Latent Action Q-Learning (LAQ). Then, the second focus of the paper is on exploring the behavior of LAQ in a variety of domains and settings. The experiments are spread across five distinct environments of different kinds (gridworlds, Atari's Freeway, and so on), and each experiment examines a different characteristic of learning.

**Summary Of The Review:**

This is a clean idea, presented carefully, with interesting theory and well conducted, insightful experiments.

---

> ### Author Response · Authors · 2021-11-20
> **Reply to reviewer Fa3L**
>
> Thank you for your thoughtful comments and feedback.
>
> **LAQ value functions allow transfer across embodiments.** Yes, it is true that in Figure 5, the confidence intervals are large and overlapping. We have increased the number of trials in our downstream task evaluations, shrinking the error bars. We tested our method against baselines using Welsh’s t-test (stronger than Student’s t-test) with a p-value of 0.05. Below are some of the results which pass this statistical test for significance:
> * On the gridworld, our method is statistically significantly faster at learning than sparse reward and BC, meaning there exists a statistically significant gap in sample means during training (where our method has converged but the sparse reward has not).
> * In the 2D continuous navigation environment, our method learns faster than densified reward with policy evaluation, BC, and sparse reward, and performs better than sparse reward and BC at convergence.
> * In Freeway, our method learns faster than densified reward with policy evaluation.
> * In the kitchen manipulation task, the strongest version of our method (using BCQ) learns faster than D3G, and sparse reward, and is superior at convergence to D3G.
> * In the embodiment transfer to quadrupedal ant experiment, our method (BCQ variant) learns faster than sparse reward.
> * In the 3D visual navigation environment, our method produces a significantly higher SPL than densified reward with policy evaluation (V_{one-act}).
>
> We have updated the plots in the paper to reflect the additional trials, and to show plus-minus standard error, to be more in line with community standards [1], and for readability. From these results, we draw two conclusions
> 1. Using densified reward for training is superior to sparse reward, and behavior cloning (statistically significantly faster or better reward at convergence on all environments, except Freeway).
> 2. The LAQ value function (trained using BCQ) is the strongest densification scheme among relevant baselines (significantly faster learning than policy evaluation on 2d continuous navigation, and superior to D3G on kitchen manipulation. Not significantly weaker than any method on any tested environment).
>
> **[Q.1]** Our methodology does not assume that the number of actions in the true environment is known. For both learning the value functions and for downstream behavior, this is not required. We use eight latent actions for all environments except the FrankaKitchen environment (for its richer underlying action space), and this number was chosen arbitrarily.
>
> **[C.1]** Excellent suggestion, we have added this to the revised paper in Section A.10.
>
> **Lemma 3.2 for stochastic policies.** Lemma 3.2 holds true for both stochastic and deterministic policies (as in this framework, deterministic policies are a subset of stochastic policies, having action probabilities of 1 or 0). We have clarified this in the appendix (A.1).
>
> Thank you for the detailed writing suggestions. We have updated the paper to reflect these changes.
>
> **References:**
>
> **[1]** ​​Henderson, Peter, et al. "Deep reinforcement learning that matters." Proceedings of the AAAI conference on artificial intelligence. Vol. 32. No. 1. 2018.

---

### Official Review · Reviewer_cKHF · 2021-11-02

**Correctness:** 3
**Technical Novelty And Significance:** 3
**Empirical Novelty And Significance:** 3
**Recommendation:** 6
**Confidence:** 4

**Main Review:**

Main points:
- I find the introduction to be well-written  and the motivation for this problem to be compelling. In particular, the idea that the same collected data could be used for different types of agents/embodiments is appealing to me, with potentially many practical use-cases.

- For the theoretical result, I find that the conditions needed to satisfy theorem 3.1 may be too stringent. If I understand correctly, it amounts to assuming that the new action set $\hat{A}$ contains all the actions of the original $A$, with possible duplicates. So, it does not allow other actions in $\hat{A}$ that are different than those already in $A$, which is very restrictive. Essentially, it seems to say that we need to obtain exactly the same actions in $A$.
I would find the theorem to be more meaningful if it allowed other actions in $\hat{A}$ at least, with perhaps a different conclusion. Adding new actions to $\hat{A}$ that are functionally different than those in $A$ may increase the optimal values in the general case, but maybe something more specific can be said under different conditions.
Another possible extension to the theorem would be to prove the necessity of the refinement assumption i.e. something like, if the optimal values in the two MDPs match, then one action space is a refinement of the other.

- The experiments were well done with approppriate baselines being chosen and a reasonable choice of environments. The variety of different evaluations to assess the method was welcome. I appreciated the experiments which tried to directly assess the quality of the learned value functions instead of only indirectly doing so with overall performance numbers.
- I liked the experiments with changing the embodiments of the agent. I haven't exactly this kind of experiment before and it seems to me that it could be applicable to certain practical scenarios.
- Planning with low-level controllers was also interesting and did a good job showcasing the strengths of the proposed approach.

Questions and suggestions:
- One aspect that confuses me is that the proposed algorithm is used to learn an action space but afterwards, this new action space is hardly used except to run Q-learning and obtain state-value functions. It seems like there should be an oppportunity to make more use of the learned action space. From an intuitive standpoint, it's not very clear to me why learning an action space is a good approach for action-less data.

- Following up on the previous point, LAQ obtains a state-value function in the end, which can be used in various ways. LAQ also learns a one-step model as an intermediate step before doing Q-learning with the new actions. Have you considered trying to learn the state-value function directly as an alternate approach? This could be done using value iteration on the state-values with the learned model.

- Did you try to interpret the learned actions at all? Do they seem to match the base actions or are they completely different?

- Can you clarify what the clustering approaches are? I did not see any explanations about how they are used.

- In Freeway, behaviour cloning seems to be performing the best. Why is that? Also, is there missing a green line extending from it for behaviour cloning + RL?

- In table 1, how exactly is the Spearman correlation computed? Is it calculated using state-value functions on a batch of data? Or is it using state-action value functions? How is the batch of data selected?

- Fig 4 plots are not clear. Is there learning in the actual environment? If that is the case, are batch RL methods used then?
Are they first used to initialize the method and then continue learning?

- In section 5, the term "state-conditioned purity" is introduced but then it isn't explained in the later sections (only in the appendix). I suggest clarifying this in the main text or moving that entire discussion to the appendix.

- The (theoretical) limitation of LAQ to deterministic MDPs is quite strong. While there's a small discussion included in the text, have you done any experiments trying it out in stochastic MDPs? Perhaps it can do reasonably well anyway.

- I would clarify that the word "refinement" in this paper is defined to be something specific and doesn't have a standard meaning. This isn't clear right now in the abstract or the introduction. I was confused since "refinement" is related to a set being a subset of another in mathematics.

**Summary Of The Paper:**

This paper focuses on the problem of learning from an offline batch of data containing only states and rewards (no actions). A theoretical result shows when we can expect to recover the true optimal value function when using a different action space. This motivates the algorithm Latent Action Q-learning (LAQ), which first learns a discrete action space by modeling one-step transitions and then applies Q-learning with the learned action space. Experiments in a variety of settings, including reward-shaping and planning with low-level controllers, validate the effectiveness of the approach.

**Summary Of The Review:**

The paper tackles an interesting topic and proposes a simple approach which performs well empirically. I don't think the theoretical result is very strong, but the paper focuses more on the empirical aspects so it's not a major concern. Also, the paper is generally well-written although there are certain areas that require more clarification, which is easily addressed.
Overall, given the thorough experiments and the interesting problem, I'm leaning towards acceptance.

---

> ### Author Response · Authors · 2021-11-20
> **Reply to reviewer cKHF (2/2)**
>
> **Details about Spearman correlation computation.** The Spearman correlation is calculated using state-value functions on a batch of data, not the state-action value function. A large batch of data (ranging from 1600 to 16000 samples, depending on environment) is randomly sampled from the dataset to compute this.
>
> **Details about Figure 4.** We clarify that our method has 3 stages:
> 1. Assign latent actions to the given action-less offline data,
> 2. Use assigned latent actions to conduct offline Q-learning, and
> 3. Use learned value functions to speed up behavior acquisition through online interaction in the environment.
>
> The plots in Figure 4 show stage 3 of this process and thus make use of interaction with an environment. The number of interaction samples are plotted on the x-axis. We use offline Q-learning for stage 2 but could equally well have used any other batch RL method (we already tried BCQ as reported in the paper). In stage 3, instead of initializing models with value functions learned in stage 2, we use them to densify the reward functions used for standard RL training in stage 3.
>
> For the baseline BC+RL line in Figure 4, BC is used as initialization before fine-tuning with RL.
>
> **State-conditioned purity mentioned without explanation.** Good point. We have included a brief definition in the main text (Section 5) as well as a more detailed definition in appendix A.6.
>
> **Experiments on stochastic MDPs.** We have conducted experiments on stochastic MDPs in the gridworld environment and have included our results in A.9 in the revised paper. Specifically, we do this by adding sticky actions, such that with a certain probability (called the stickiness parameter), the environment executes the previous action as opposed to the given action. We see that as the probability of taking the previous action (i.e. the stickiness parameter) increases, the state-based purity of the learned latent actions decreases. We also see that the spearman’s rho decreases, but not by a significant margin, even when the stickiness parameter is set to 95%. With the spearman’s rho being high, even for highly stochastic MDPs, we see that the proportion of correct actions is 100% for all of the different stochasticity levels. However, it is possible to also construct simple scenarios where LAQ, and for that matter any deterministic algorithm, will suffer as we describe in Section A.2.
>
> **Definition of refinement.** This is a good point. We have updated the paper to include a definition to this effect in Section 3.

---

> ### Author Response · Authors · 2021-11-20
> **Reply to reviewer cKHF (1/2)**
>
> Thank you for your thoughtful comments and feedback.
>
> **Extension to Theorem 1.** Thank you for the suggestion. It certainly seems possible that one could place a bound on the error in the induced value function based on divergences in the action space. We view that as beyond the scope of this paper, and a promising area of future work. As for the necessity of refinement, one can prove the non-necessity of refinement by counterexample. Adding a new action to the MDP with unique transition dynamics, but which would never be taken by the optimal policy makes the new action space not a refinement, while still having the same value function.
>
> We respond to the specific questions and suggestions below:
>
> **Why is learning an action space a good approach for action-less data?** Our intuition is that in the setting where there are no action labels, we need a way to differentiate between different transitions from the same state. Not doing so amounts to TD(0) policy evaluation (and not Q-learning). TD(0) learns the value of the policy that was used to collect the dataset, which could be sub-optimal. Differentiating between transitions using latent actions enables the use of Q-learning and allows for learning the optimal value function. Empirically, we observe improvements over TD(0) and other baselines across different metrics and different environments. An alternative is D3G, which does better than TD(0), but does not work for high-dimensional state spaces and comes short when directly compared to our method, LAQ.
>
> As we try to demonstrate in the Appendix (A.6), the purity of the learned latent actions with respect to the ground truth environment actions does not have to be high, but rather the state-based purity needs to be high to learn good value functions. Direct alignment of the learned latent action space with the ground truth environment action space is difficult (see analysis for Freeway in A.7, the same latent action maps to different ground truth actions in different parts of space), and as we show (our definition of refinement only requires a per-state correspondence between actions), unnecessary. By using Q-learning, we side-step this issue and allow for learning good value functions as long as the state-based purity is high.
>
> **Learned action space is hardly used.** We’d be happy to try any suggestions as to how else the learned action space could be utilized the reviewer may have. But, as we note above, the latent action space may not be aligned to the ground truth action space, which may make direct utilization difficult.
>
> **Using value iteration on the state-values with the learned model as an alternate approach.** Value iteration is in fact an alternative way of viewing our approach, with some differences. First, because in environments with continuous state spaces like FrankaKitchen, we cannot enumerate over all states, we sample quadruples from the dataset for training. Second, using the learned forward model would produce imperfect predictions of the next states for value iteration, particularly for high dimensional observation spaces. Our use of Q-learning which uses samples from the dataset for training sidesteps both of these issues.
>
> **Interpretation of latent actions.** In the Appendix (A.7), we visualize the learned latent actions for Freeway. Figure S.9 qualitatively shows what transitions get mapped to what latent actions. Interestingly, the same latent action takes on a different meaning in different parts of the state space. Thus, the overall purity is not high, but the state-based purity of the learned latent actions is high. That is, given the state and the latent action, you can predict the ground truth action with high certainty. Even though the latent actions may not directly correspond to the base action space, this does not pose a problem as we prove that only a state-based refinement is necessary to recover the optimal value function.
>
> **Clarification on clustering approaches.** We include two clustering baselines. Clustering concatenated observations clusters consecutive observations $[o_{t+1}; o_t]$, and clustering observation differences (Diff) clusters the difference between the two consecutive observations $[o_{t+1} - o_t]$. We used k-means clustering using the scikit implementation, which automatically does 10 clustering trials and picks the best clustering based on the loss.
>
> **Behavior cloning on Freeway.** In Freeway, the dataset largely consists of successful episodes where the policy used to generate the data can completely solve the game. For this reason, behavior cloning does very well. We did not compute the missing green behavior cloning + RL line for Freeway as behavior cloning by itself solves the task -- the maximum reward in the environment is 30, which is achieved by behavior cloning.

---

### Official Review · Reviewer_YoH4 · 2021-11-03

**Correctness:** 3
**Technical Novelty And Significance:** 3
**Empirical Novelty And Significance:** 2
**Recommendation:** 6
**Confidence:** 4

**Main Review:**


Strengths
---------

-   An interesting approach to the problem of learning from action-less
    experience. The three part procedure of estimating latent actions
    with EM, doing Q-learning with the learned actions and using the
    value function on downstream tasks is promising and seems new.

-   The theoretical arguments that underpin the proposed method are
    mostly well thought out. There are some confusing parts, that I
    detail below, but I do think the overall approach is correct and the
    theory cleanly motivates the proposed algorithm.

Weaknesses
----------

-   The significance of the contribution is unclear. This is in part due
    to the obscure nature of the problem (of learning with action-less
    experience), partly because the related work is not well
    contextualized earlier in the paper and partly due to the
    experimental evaluation.

-   The experiments, while somewhat comprehensive, are not particularly
    compelling. The results are mostly evaluated using correlation of
    the learned value function and on downstream tasks, both of which
    makes assessing the overall effectiveness difficult without
    extensive ablation. It is not clear to me that correlation is a good
    performance criterion in this case. Furthermore, the results on
    downstream tasks are mostly inconclusive with overlapping confidence
    intervals.

Detailed Comments
-----------------

-   In Section 1 Paragraph 2, you motivate the method by stating that
    the learned value function can be helpful in many downstream tasks,
    including control. This is true, but the state-value function can be
    learned without actions already. The paragraph does not communicate
    the advantage of learning the action-value function with imputed
    actions instead of the state-value function.

-   Section 1, Paragraph 3 "Thus, the central technical question is how
    to learn a good value function from undirected observation streams."

    Sentences such as the one above should specify "action-value"
    function, because, as I mentioned above, a state-value function can
    be directly learned. This is partially addressed in the rest of
    paragraph 3 and in Section 3, but to say that Q-learning without
    actions is policy evaluation seems incorrect. Q-learning cannot be
    applied without actions, unless you impute the actions. A
    state-value function can still be learned using TD learning, which
    seems like the point that is being made in this paragraph.

-   Section 3, Regarding the choice of 4x - ground truth. Is balancing
    the actions in the larger set important? If one action is repeated
    15 times and the other ground truth actions are only repeated 1
    times, would that make the problem harder? Another more pressing
    issue is if the actions are not a refinement, such as if only two
    latent actions are permitted. This is more relevant in practice,
    where we may not know the right action space apriori.

-   "For downstream decision making, we only care about

the relative ordering of state values."

I don't understand this statement. For downstream tasks the relative
ordering of the action-values is the pertinent quantity, but I don't
think an ordering of state-values is meaningful. Since this is the
primary reason for using spearman's rank correlation, I do not see how
measuring correlation would be a useful measure of success.

-   How is correlation being measured exactly? And is this truly the
    measure of success? If correlation is used to overcome some constant
    bias in the state-values, then this implicitly assumes that the bias
    is uniform across states which may not be the case. Learning
    action-value functions offline leads to many issues in the quality
    of the value predictor which may not manifest itself in the control
    policy (such as divergent action-values for out-of-sample or
    out-of-distribution actions).

-   Ground truth actions are said to be an upper bound on the
    performance of your method, but in Table 1 for Kitchen manipulation
    the proposed method outperforms this upper bound. How is this to be
    interpreted in light of the stability comments in footnote 2? Based
    on footnote 2, it seems like the value function should be constant
    and not change between runs (i.e. shouldn't the correlation should
    be 1?).

-   The results on downstream tasks would benefit from TD0 as a
    baseline. While the value function learned is the value of the
    behavior policy (and not the optimal value function), it could still
    be used for all three settings (densified, embodiment transfer and
    guiding low level controllers).

-   Figure 4: while this does demonstrate improvement in the grid world
    setting, there is no significant difference between the methods in
    the other 2d navigation or freeway. Also, the difference in
    asymptotic performance in gridworld is strange. Shouldn't all
    methods eventually reach the optimal policy?

-   Putting the related work at the end of the paper prevents
    much-needed context for the proposed methods. The introduction does
    a good job of stating what you are doing, but the question of why is
    less clear.

Minor Comments
--------------

-   Page 4 , Line 3: ammounts -\> amounts

**Summary Of The Paper:**

This paper analyzes the problem of learning from experience tuples that
omit the action. The authors analyze tabular Q-learning and show that if
the action-space is a refinement, then Q-learning can learn the optimal
value function. Motivated by this, an algorithm is proposed that labels
$(s,s')$ tuples with a discrete action $\hat a$, with Q-learning used
with these imputed actions. The proposed algorithm is shown to surpass
several baselines on grid world, continuous control, freeway (atari),
robotic manipulation and rich visual navigation environments.

**Summary Of The Review:**


This is an interesting paper with good ideas, but is held back by an
overall confusing experimental evaluation. The theoretical arguments
that motivate the proposed algorithm are well thought out. The empirical
analysis, however, does not give confidence that the proposed algorithm
is an improvement in the action-less regime. I would like to see an
environment that clearly shows that densified rewards with the proposed
algorithm is superior to densified rewards with a learned value function
via TD0. I also think that the correlation performance criterion needs
to be better justified. I will currently leave my rating at below the
acceptance threshold, but I am open to increasing my score if some of my
concerns are addressed.

Edit: After discussion with the authors, I have increased my score to a 6.

---

> ### Author Response · Authors · 2021-11-20
> **Reply to reviewer YoH4 (2/2)**
>
> **State-value ordering:** When deriving behavior (Section 4.3) we are not taking the action-value function and acting greedily according to the Q-values, but rather using the state-value function as reward shaping for training a reinforcement learning agent. This is why the ordering of state-values is meaningful. If the value state-value function correlates highly with the ground-truth state-value function, it serves to densify the reward of the RL task, which would have sparse reward otherwise.
>
> **Spearman’s Correlations:** We would note that we are not computing a standard (Pearson) correlation, but rather the Spearman’s rank correlation. This compares the extent to which the metrics induce the same ordering of states, and would be impacted by a high variance state-dependant bias in the evaluated samples. We are computing the spearman’s rank correlation coefficient of the learned state-value function against a ground truth state-value function over a large batch of states (ranging from 1600 to 16000 samples, depending on environment) sampled randomly from the offline dataset used for latent action learning.
>
> As you point out, learning value functions offline can have issues with out-of-distribution samples, as such, the correlations are not directly a measure of success but rather an indicator that the state-value function learned from latent actions is capturing meaningful information about the underlying task. The most direct measure of success is the improved performance in downstream tasks shown in figures 4,5,6. These results indicate that what issues arise from offline value learning do not prevent our method from outperforming relevant baselines.
>
> **Asymptotic Gridworld Performance:** Here we are plotting normalized discounted return. All solutions reach the goal, however sparse reward and d3g settled into slightly sub-optimal solutions due to the poorer reward shaping. If run long enough, runs with these other shapings should eventually randomly discover the optimal solution. As for BC+RL, to finetune the BC policy we must use policy gradient instead of DQN, which is far less sample efficient and converges far slower. We do not claim that our method leads to better asymptotic performance in the gridworld, rather we want to highlight that the proper shaping leads to far faster learning than sparse reward or behavior cloning.
>
> **Ground truth actions (Table 1 - Kitchen):** The “ground truth actions” for Kitchen Manipulation in table 1 is the Spearman's rank correlation of offline DDPG with ground truth actions against our handcrafted ground-truth value function. As you suggest, the correlation of the handcrafted ground-truth value function against itself is indeed 1 across runs. This indicates that our method is on-par with offline DDPG with ground truth actions (0.905 vs 0.901) for this task.

---

> > ### Comment · Reviewer_YoH4 · 2021-11-21
> > **Thank you for your response**
> >
> > Thank you for your response! You've acknowledged and addressed many of my points, and I appreciate that you included more runs. The updated draft also clarifies many of the points that were at first confusing. One thing that remains unclear, however, is the state-value ordering. What is this order with respect to, the underlying states? I do not see how the state-values themselves could be ordered. The quote:
> >
> > "If the value state-value function correlates highly with the ground-truth state-value function, it serves to densify the reward of the RL task, which would have sparse reward otherwise."
> >
> > makes sense, but the connection to state ordering is still lost on me. Because of this, I am not able to properly interpret your use of Spearman's rank correlation. Could you give some further intuition / example for this?

---

> > > ### Author Response · Authors · 2021-11-21
> > > **Reply to reviewer YoH4**
> > >
> > > Thank you for your prompt response!
> > >
> > > The Spearman’s rank correlation we compute assesses if two different scalar output functions induce the same relative order of inputs. We have two value functions, one which is considered the reference value function V_ref, and one which we wish to test against it (learned using our method or baseline) V_x. For all environments except the kitchen we use a value function learned with offline DDPG with ground-truth actions as the ground-truth value function (see section 5.1). For the kitchen we use a hand-defined value function as the ground-truth value function (see footnote 3). We compute V_ref(s), and V_x(s) for a fixed set of states. For completeness, we include pseudocode for how this is implemented in our experiments:
> > >
> > >     S ← sample an arbitrarily ordered set of states from the offline dataset D
> > >     Initialize empty lists V_xs and V_refs
> > >     for i from 0 to len(S)-1:
> > >         V_xs.append(V_x(S[i]))
> > >         V_refs.append(V_ref(S[i]))
> > >     endfor
> > >     correlation = scipy.stats.spearmanr(V_xs, V_refs)
> > >
> > > The Spearman’s rank correlation is high when the ordering of states induced by the value functions is similar.
> > >
> > > To be more concrete, let’s say we have $5$ states $S = [s1,s2,s3,s4,s5]$, and the reference values for these states are V_refs $= [1,2,3,4,5]$. If V_x produced the following values for those states, they would all have Spearman’s rank correlation of 1: V_xs $ = [1,2,3,4,5]$ or $[1,2,3,4,100]$ or $[-10,0,7,8,9]$. This is because they all give the samples the same relative ordering. If V_x instead produced the values V_xs $= [1,2,3,4,0]$, this would have a lower Spearman’s rank correlation, because state $s5$ was ranked the highest according to the ground truth, but not by V_x.
> > >
> > > Hopefully this makes things clearer. We would be happy to take any further clarifying questions the reviewer may have.

---

> > > > ### Comment · Reviewer_YoH4 · 2021-11-23
> > > > **Thanks for the example.**
> > > >
> > > > Thank you, the explanation and example was very helpful in understanding how and why you use spearman's rank correlation. My interpretation: if the state-value V_x is relatively ordered with respect to an accurate reference value $V_{ref} \approx V^*$, then (approximate) dynamic programming with V_x for reward shaping will help because one state lookahead will exactly recover the optimal policy. I believe this may be a common stumbling point for other readers, so please consider including some explanation/justification for the use of spearman's rank correlaiton in the camera-ready if accepted.

---

> ### Author Response · Authors · 2021-11-20
> **Reply to reviewer YoH4 (1/2)**
>
> Thank you for your thoughtful response to our paper. We’ve incorporated your feedback into an updated version of the paper. In this response, we will describe the changes we’ve made as well as directly address some specific concerns/questions. Hopefully it allows you to make a more informed judgment of our paper.
>
> **Related Work:** We appreciate your feedback that our introduction should include further discussion of related work. In our updated draft we have included additional language to this effect. However, we disagree with the characterization of the problem of learning with action-less experience as “obscure”. To this end, we note that many papers have been published in major conferences over the last four years tackling this problem. We cite many of these in related work (Section 7 - “State-only Learning”).
>
> **Experimental Significance:** We have increased the number of trials in our downstream task evaluations, shrinking the error bars. We tested our method against baselines using Welsh’s t-test (stronger than Student’s t-test) with a p-value of 0.05. Below are some of the results which pass this statistical test for significance:
>   - On the gridworld, our method is statistically significantly faster at learning than sparse reward and BC, meaning there exists a statistically significant gap in sample means during training (where our method has converged but the sparse reward has not).
>   - In the 2D continuous navigation environment, our method learns faster than densified reward with policy evaluation, BC, and sparse reward, and performs better than sparse reward and BC at convergence.
>   - In Freeway, our method learns faster than densified reward with policy evaluation.
>   - In the kitchen manipulation task, the strongest version of our method (using BCQ) learns faster than D3G, and sparse reward, and is superior at convergence to D3G.
>   - In the embodiment transfer to quadrupedal ant experiment, our method (BCQ variant) learns faster than sparse reward.
>   - In the 3D visual navigation environment, our method produces a significantly higher SPL than densified reward with policy evaluation (V_{one-act}).
>
> We have updated the plots in the paper to reflect the additional trials, and to show plus-minus standard error, to be more in line with community standards [1], and for readability. From these results, we draw two conclusions
> 1. Using densified reward for training is superior to sparse reward, and behavior cloning (statistically significantly faster or better reward at convergence on all environments, except Freeway).
> 2. The LAQ value function (trained using BCQ) is the strongest densification scheme among relevant baselines (significantly faster learning than policy evaluation on 2d continuous navigation, and superior to D3G on kitchen manipulation. Not significantly weaker than any method on any tested environment).
>
> **State-value vs Action-value:** As you point out, a state-value function can be learned directly without action information. Naively, this amounts to TD(0) policy evaluation, which we discuss briefly in section 3. In our original draft, we compare against
> TD(0) in terms of spearman’s correlation (Table 1 - “Single Action”), and directly in the 3D visual navigation environment.
> The for other downstream tasks we compared against the more sophisticated state-value estimation method of D3G. In these evaluations, our method is superior.
>
> We have now updated the draft to include a direct comparison against TD(0) for downstream tasks, and have updated the language in the intro to indicate that our method is not the only way one could derive state-value functions (in paragraph 4).
>
> Our goal is indeed to learn a good state-value function, we simply use an action-value function as an intermediate representation to arrive at a good state-value function, since learning directly with policy evaluation is sub-optimal.
>
> You are right that Q-learning requires actions, when we say “Q-learning without actions” we are implicitly assuming the most naive imputation of actions (all state transitions occur from one action class). We have updated the language in the paper to make this less confusing.
>
> **Refinement Distribution/Non-refinement:** In the tabular case the distribution of the refinement should not matter at convergence as long as the optimal state transitions appear in the offline dataset.
>
> With regards to non-refinement action spaces: all continuous action experiments address this question. Since we model our latent action spaces as discrete when fitting a dataset with continuous actions a true refinement cannot be learned. However, our method is still able to perform well in all continuous action experiments.
>
> **References:**
>
> **[1]** ​​Henderson, Peter, et al. "Deep reinforcement learning that matters." Proceedings of the AAAI conference on artificial intelligence. Vol. 32. No. 1. 2018.

---

### Author Response · Authors · 2021-11-19
**Updated Draft**

We thank the reviewers for their thoughtful comments. We have incorporated their feedback to produce an updated version of the draft, with additional experiments, which we believe addresses all concerns raised. We respond to the individual questions and concerns of each reviewer in the individual responses, providing clarification and pointers to material in the paper.

The updated draft reflects the following changes:
- Updated downstream RL results with more trials
- New discussion in related work on latent MDP learning
- Clarifications / presentation changes:
  - Refined motivating language in introduction
  - Motivated using learned action space over TD(0) policy evaluation in the introduction
  - Contextualized our method with related work in the introduction
  - Added the definition of value function in preliminaries
  - Factored out the definition of refinement in our theoretical framework
  - Added brief sketch/intuition about the proof of lemma 3.2 in the main paper
  - Defined state-conditioned purity in the main text

---

### Decision · Program_Chairs · 2022-01-20

**Decision:**

Accept (Poster)

**Comment:**

The paper proposes a method for learning state value functions from (s,s',r) tuples, founded on the theoretical analysis in MDP setting. The extensive evaluation in several environments shows the benefit of the algorithm.

The consensus among the reviewers, and I concur, that the paper proposes an interesting and novel method. It is cleanly presented, and well-founded. The evaluations across range of environments, including robot manipulation validate the method.

During the rebuttal, the authors provided additional evaluation, added a discussion on the latent MDPs, and made numerous clarification, addressing most / all reviewers' questions.